# Early learning of the optimal constant solution in neural networks and humans

**Jirko Rubruck (jirko.rubruck@stx.ox.ac.uk)**
Experimental Psychology, University of Oxford

**Jan P. Bauer (jan.bauer@ucl.ac.uk)**
Gatsby Unit, University College London & ELSC, Hebrew University

**Andrew Saxe**[†]
Gatsby Unit, University College London

**Christopher Summerfield**[†]
Experimental Psychology, University of Oxford

† Co-senior authors

## Abstract

**Deep neural networks learn increasingly complex functions over the course of training. Here, we show both empirically and theoretically that learning of the target function is preceded by an early phase in which networks learn the optimal constant solution (OCS) – that is, initial model responses mirror the distribution of target labels, while entirely ignoring information provided in the input. Using a hierarchical category learning task, we derive exact solutions for learning dynamics in deep linear networks trained with bias terms. Even when initialized to zero, this simple architectural feature induces substantial changes in early dynamics. We identify hallmarks of this early OCS phase and illustrate how these signatures are observed in deep linear networks and larger, nonlinear convolutional neural networks solving a hierarchical learning task based on MNIST and CIFAR10. We train human learners over the course of three days on a structurally equivalent learning task. We then identify qualitative signatures of this early OCS phase in terms of true negative rates. Surprisingly, we find the same early reliance on the OCS in the behavior of human learners. Finally, we show that learning of the OCS can emerge even in the absence of bias terms and is equivalently driven by generic correlations in the input data. Overall, our work suggests the OCS is a common phenomenon in biological and artificial, supervised, error-corrective learning, and suggests possible factors for its prevalence.**

## Introduction

Neural networks trained with stochastic gradient descent (SGD) exhibit various *simplicity biases*, where models tend to learn simple functions before more complex ones (Kalimeris et al., 2019; Rahaman et al., 2019; Hu et al., 2020; Cao, Fang, et al., 2020). Simplicity biases hold significant theoretical interest as they provide an explanation for how deep networks generalize or fail to generalize in practice (Bhattamishra et al., 2023; Valle-Pérez et al., 2019; Zhang et al., 2021).

The characterization of simplicity biases is still incomplete. Some explanations appeal to distributional properties of input

data, pointing out that SGD progressively learns increasingly higher-order moments (Refinetti et al., 2023; Belrose et al., 2024). Other approaches focus directly on the evolution of the network function, proposing that networks initially learn a classifier highly correlated with a linear model. Importantly, networks continue to perform well on examples correctly classified by this simple function, even when overfitting in later training (Kalimeris et al., 2019). This implies that dynamical simplicity biases help models generalize, by locking in initial knowledge that is not erased or forgotten during later training (Braun et al., 2022; Kalimeris et al., 2019).

Deep linear networks have proven to be a valuable tool for studying simplicity biases. A key finding is that directions in the network function are learned in order of importance (Saxe et al., 2014, 2019). This phenomenon, known as *progressive differentiation*, connects modern deep learning theory to both to human development and to the earliest connectionist models of semantic cognition (Rogers & McClelland, 2004; Rumelhart et al., 1986).

Our contribution proposes a connection between these works by characterizing networks in the *earliest* stages of learning in terms of input biases, output statistics, and architecture. In the hierarchical setting by Saxe et al. (2019), we demonstrate both theoretically and empirically that neural networks initially learn via the output statistics of the data. This function has been termed the optimal constant solution (OCS) by Kang et al. (2024), who demonstrated that networks revert to the OCS when probed on out-of-distribution inputs. Here, we demonstrate and prove how linear networks, when equipped with bias terms, necessarily learn the OCS early in training. Fig. 1A graphically illustrates this observation. We furthermore highlight the practical relevance of these results by examining early learning dynamics in complex, non-linear architectures.

Biological learners also display behaviors that imply the input-independent learning of output statistics. In probability matching, responses mirror the probabilities of rewarded actions (Herrnstein, 1961; Estes, 1964; Estes & Straughan, 1954). Learners often display non-stationary biases that are driven by the distribution of recent responses (Jones et al., 2015; Gold et al., 2008; Verplanck et al., 1952). In paired-associates learn-

ing accuracy can depend not only on a learned input-output mapping but also on knowledge of the task structure (Hawker, 1964; Bower, 1962). Humans also display simplicity biases and preferentially use simple over complex functions (J. Feldman, 2000; Goodman et al., 2008; Chater, 1996; Lombrozo, 2007; J. Feldman, 2003). However, relatively little attention has been devoted to the dynamics of these biases. We conduct experiments to determine whether humans replicate *early* reliance on the OCS.

## Contributions

- We devise exact solutions for learning dynamics to analyze linear networks with bias terms in the input layer. Even when initialized at zero, this component substantially alters *early* learning dynamics.

- We empirically characterize early learning in these linear networks as dominated by average output statistics. We explain this result with a theoretical analysis which reveals that average output statistics are always learned first when the network contains bias terms.

- We further highlight the practical relevance of these results for humans in a hierarchical learning task. We empirically demonstrate that learners develop similar stereotypical response biases during the early stages of training.

- On the basis of the developed theory, we show that, in linear networks, early OCS learning can be induced by input correlations even in the absence of bias terms. For natural datasets, we empirically demonstrate that OCS learning can be purely driven by generic correlations in the input data.

## Related work

**Deep linear networks.** In deep linear networks analytical solutions have been obtained for certain initial conditions and datasets (Saxe et al., 2014, 2019; Braun et al., 2022; Fukumizu, 1998). Progress has also been made in understanding linear network loss landscapes (Baldi & Hornik, 1989) and generalization ability (Lampinen & Ganguli, 2019). Despite their linearity these models display complex non-linear learning dynamics which reflect phenomena observed in non-linear models (Saxe et al., 2019). Moreover, learning dynamics in such simple models have been argued to qualitatively resemble phenomena observed in the cognitive development of humans (Saxe et al., 2019; Rogers & McClelland, 2004). Here, we consider more general architectures and data, and connect this framework to experiments.

**Biological response biases.** Humans and animals routinely display response biases during perceptual learning and decision making tasks (Gold et al., 2008; Jones et al., 2015; Liebana Garcia et al., 2023; Amitay et al., 2014; Urai et al., 2019). In these tasks decisions are frequently made in sequences where responses and feedback steer decisions beyond the provided perceptual evidence (Jones et al., 2015; Fan et al., 2024; Gold et al., 2008; Verplanck et al., 1952; Sugrue et al., 2004). Non-stationary response biases can be driven by feedback on previous trials (Dutilh et al., 2012; Rabbitt & Rodgers, 1977) or might reflect global beliefs about the statistics of a task (Fan et al., 2024; Jones et al., 2015). Importantly, response biases are particularly pronounced in early learning (Jones et al., 2015; Gold et al., 2008; Liebana Garcia et al., 2023) and their influence appears to be strongest when uncertainty about the correct response is highest (Gold et al., 2008; Fan et al., 2024). We here develop a neural network model to study the mechanism behind this phenomenon.

**Simplicity biases in machine learning.** Simplicity biases in neural networks have been studied extensively both theoretically (Bordelon et al., 2020; Mei et al., 2022) and empirically (Bhattamishra et al., 2023; Mingard et al., 2023). Work on the *distributional* simplicity bias emphasizes the importance of input data and proposes that models learn via progressive exploitation of dataset moments (Refinetti et al., 2023; Belrose et al., 2024). On the other hand, neural networks have been found to express simpler functions during early training (Kalimeris et al., 2019; Refinetti et al., 2023) where the dynamics are dependent on their NTK spectrum (Rahaman et al., 2019; Cao, Summerfield, & Saxe, 2020). Further, for certain well-behaved input-distributions non-linear neural networks have been found to behave like a linear model on the data during early training (Hu et al., 2020). Our work supplements these results and focuses on the particular phenomenon of OCS learning in the early training.

## Paper organization

We initially review the linear network formalism on which we base our theoretical analysis. We then derive learning dynamics for linear networks with bias terms trained on a classic hierarchical task and we document substantial changes in early dynamics. The following section characterizes this period of early learning empirically, and provides a theoretical explanation. We then validate the relevance of our findings for learning in complex models. Subsequently, we demonstrates the prevalence of early OCS learning in humans. Finally, we probe generality by considering natural datasets and models that do not strictly fulfill the previous theoretical assumptions.

## Linear network preliminaries

Here, we briefly review the analytical approach to learning dynamics in linear networks developed by Saxe et al. (2014, 2019). Consider a learning task in which a network is presented with input vectors $\mathbf{x}_i \in \mathbb{R}^{N_{in}}$ that are associated to output vectors $\mathbf{y}_i \in \mathbb{R}^{N_{out}}$. The total dataset consists of $\{\mathbf{x}_i, \mathbf{y}_i\}_{i=1}^{N}$ with $N$ samples. For our setting we consider two layer linear networks where the forward pass computes $\hat{\mathbf{y}}_i = \mathbf{W}^2 \mathbf{W}^1 \mathbf{x}_i$ and shallow networks with forward pass $\hat{\mathbf{y}}_i = \mathbf{W}^s \mathbf{x}_i$. Here weight matrices are of dimension $\mathbf{W}^1 \in \mathbb{R}^{N_{hid} \times N_{in}}$, $\mathbf{W}^2 \in \mathbb{R}^{N_{out} \times N_{hid}}$, and $\mathbf{W}^s \in \mathbb{R}^{N_{out} \times N_{in}}$. We train our networks to minimize a squared error loss of the form $\mathcal{L}(\hat{\mathbf{y}}) = \frac{1}{2} \sum_{i=1}^{N} \|\mathbf{y}_i - \hat{\mathbf{y}}_i\|^2$.

We optimize networks using full batch-gradient descent in the gradient flow regime. When learning from small initial conditions, dynamics in these simple networks are solely dependent on the dataset input-output and input-input correlation matrices

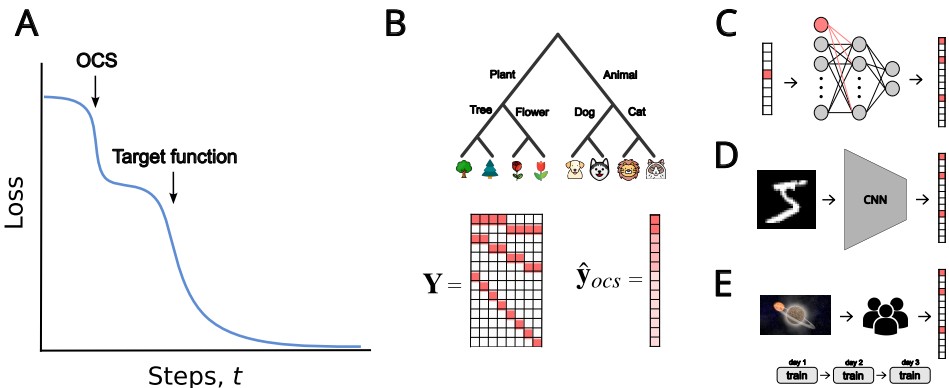

Figure 1: Early learning of the optimal constant solution (OCS). **A** Graphical illustration of our hypothesis, where learning of the target function is preceded by the early acquisition of the OCS. **B** *Top*: a graphical illustration of the hierarchical structure embedded in the outputs. *Bottom*: The full output data matrix $\mathbf{Y}$ used across different types of learners and the corresponding OCS solution $\hat{\mathbf{y}}_{ocs}$. **C,D,E** Illustration of experiments in linear networks with bias terms, non-linear models, and the task as adapted for humans.

(Saxe et al., 2014). Using singular value decomposition (SVD), these matrices can be expressed as

$$\boldsymbol{\Sigma}^{yx} = \frac{1}{N}\mathbf{Y}\mathbf{X}^T = \mathbf{U}\mathbf{S}\mathbf{V}^T, \quad \boldsymbol{\Sigma}^x = \frac{1}{N}\mathbf{X}\mathbf{X}^T = \mathbf{V}\mathbf{D}\mathbf{V}^T. \quad (1)$$

Here $\mathbf{X} \in \mathbb{R}^{N_{in} \times N}$ and $\mathbf{Y} \in \mathbb{R}^{N_{out} \times N}$ contain the full set of input vectors and output vectors. Crucially, if the right singular vectors $\mathbf{V}^T$ of $\boldsymbol{\Sigma}^{yx}$ diagonalize $\boldsymbol{\Sigma}^x$ (see Proposition 1) then the full evolution of network weights for deep and shallow networks through time can be described as

$$\mathbf{W}^2(t)\mathbf{W}^1(t) = \mathbf{U}\mathbf{A}(t)\mathbf{V}^T. \quad (2)$$

Here $\mathbf{A}(t)$ is a diagonal matrix. The evolution of these diagonal values $\mathbf{A}(t)_{\alpha\alpha} = a_\alpha(t)$ at each time-step $t$ then follows a sigmoidal trajectory as expressed in Eq. (3). For shallow networks we can similarly describe the evolution of the weight matrix $\mathbf{W}^s(t)$ as $\mathbf{U}\mathbf{B}(t)\mathbf{V}^T$. Here diagonal values $\mathbf{B}(t)_{\alpha\alpha} = b_\alpha(t)$ evolve as seen in Eq. (4)

$$a_\alpha(t) = \frac{s_\alpha/d_\alpha}{1 - (1 - \frac{s_\alpha}{d_\alpha a_0})e^{-\frac{2s_\alpha}{\tau}t}} \quad (3)$$

$$b_\alpha(t) = \frac{s_\alpha}{d_\alpha}(1 - e^{-\frac{d_\alpha}{\tau}t}) + b_0 e^{-\frac{d_\alpha}{\tau}t} \quad (4)$$

In Eq. (3) $s_\alpha = \mathbf{S}_{\alpha\alpha}$ and $d_\alpha = \mathbf{D}_{\alpha\alpha}$ denote the relevant singular values of $\boldsymbol{\Sigma}^{yx}$ and the eigenvalues of $\boldsymbol{\Sigma}^x$ respectively, $a_0$ are the singular values at initialization, and $\tau = \frac{1}{N\varepsilon}$ is the time constant with $\varepsilon$ the learning rate. In Eq. (4) $b_0$ is the initial condition given by the initialization. These relations reveal that singular values control learning speed. The solutions hinge on the diagonalization of $\boldsymbol{\Sigma}^x$ through $\mathbf{V}$. Much prior work focused on the case of white inputs, i.e. $\boldsymbol{\Sigma}^x = \mathbf{I}_N$ where $\mathbf{I}_N$ denotes the $N \times N$ identity matrix. The solution holds trivially as any $\mathbf{V}$ will orthogonalize $\boldsymbol{\Sigma}^x$. We discuss a relevant relaxation of this condition in Proposition 1. While solutions can be derived for

non-white inputs as in auto-encoding problems (Benjamin et al., 2022; Saxe et al., 2019), less attention has been devoted to these dynamics. We will show how these solutions apply when networks contain bias terms in the input layer.

## Exact learning dynamics with bias terms

In this section, we derive exact learning dynamics in linear networks with bias terms and analyse the resulting changes in the dynamics. This extension to the theory by Saxe et al. (2014) forms the basis for our later discussion. For simplicity, we focus on input bias terms and uncorrelated data, but explore bias terms in other layers and correlated inputs in the Appendix (*Learning dynamics for bias terms*) and our later section on generic input correlations, respectively.

**Closed-form learning dynamics.** We consider uncorrelated inputs $\mathbf{X} = \mathbf{I}_N$ where $\mathbf{I}_N$ denotes the $N \times N$ identity matrix. Our linear network with a bias term in the input layer will compute $\mathbf{W}^2(\tilde{\mathbf{W}}^1\mathbf{x}_i + \tilde{\mathbf{b}})$ where $\tilde{\mathbf{b}}$ are learnable bias terms.

A priori, it is unclear whether the diagonalisation of of $\boldsymbol{\Sigma}^x$ through $\mathbf{V}$ in Eq. (1) is possible in presence of bias terms. Here, we state the condition under which learning dynamics can be described in closed-form.

**Proposition 1** (Feasibility of closed-form learning dynamics). *For any input data $\mathbf{X} \in \mathbb{R}^{N_{in} \times N}$ and output data $\mathbf{Y} \in \mathbb{R}^{N_{out} \times N}$ it is possible to diagonalize $\boldsymbol{\Sigma}^x$ by the right singular vectors $\mathbf{V}$ of $\boldsymbol{\Sigma}^{yx}$ if $\mathbf{Y}^T\mathbf{Y}$ and $\mathbf{X}^T\mathbf{X}$ commute. The converse holds true only if $\mathbf{X}$ has a left inverse.*

A proof is given in the Appendix (*Feasibility of closed-form solution*). We put this statement to use to assess the effect of a bias term on learning, building on the linear network formalism. To this end, we re-express the network weights as $\mathbf{W}^1 = \begin{bmatrix} \tilde{\mathbf{b}} & \tilde{\mathbf{W}}^1 \end{bmatrix}$ with inputs defined as $\mathbf{x}_i = \begin{bmatrix} 1 & \mathbf{I}_i^T \end{bmatrix}^T$ where $\mathbf{I}_i$ denotes the $i$th column of the $N \times N$ identity matrix (see also Appendix *Equivalence of bias terms*). To introduce a controlled

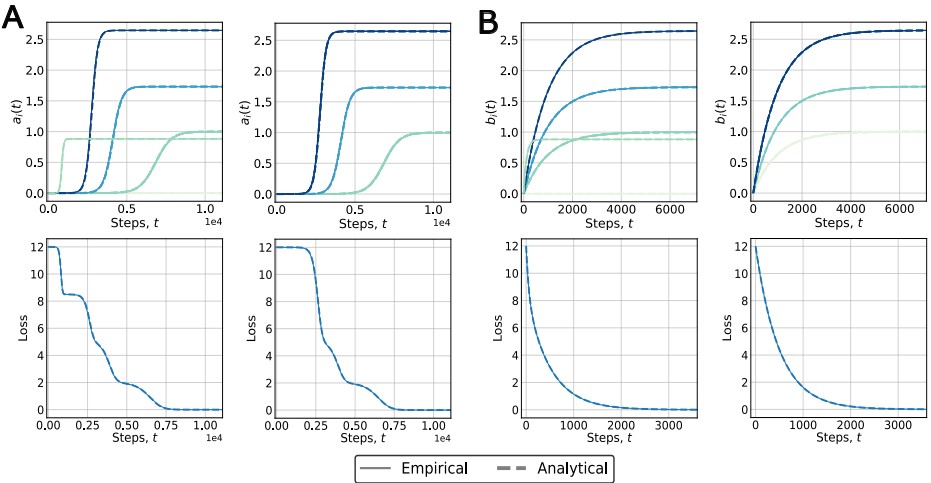

Figure 2: Exact learning dynamics. **A** Deep linear networks with bias (left) and without bias term (right). **B** Shallow linear networks with bias (left) and without bias term (right). *Top row:* Exact and simulated effective singular values $\mathbf{A}(t)$ and $\mathbf{B}(t)$ for deep and shallow linear networks respectively. Different $a_\alpha(t)$ and $b_\alpha(t)$ are color-coded according to their asymptote value with larger values as darker. *Bottom row:* Exact and simulated loss.

setting in which to analyze the effect of bias terms, we now first consider a canonical hierarchical learning task. Later sections of the paper will generalize our findings beyond this setting.

**The hierarchical task.** The hierarchical task requires learning a mapping from one-hot, input vectors to output vectors that are depicted in Fig. 1B. Hereby each output vector is "three-hot", i.e. the vector has three entries/labels. The hierarchical structure arises from the similarity between output vectors where some labels $y^m(\mathbf{x}_i)$ are more general and correspond to more than one input $\mathbf{x}_i$, while labels corresponding to the bottom of the hierarchy are specific to a single input vector $\mathbf{x}_i$. The task is motivated in the literature on semantic cognition and leverages the fact that semantic information is usually hierarchically structured (Rogers & McClelland, 2004). In Fig. 2 we depict exact learning trajectories for the hierarchically structured outputs from Fig. 1B.

Importantly, the introduction of a bias term $\mathbf{X} \rightarrow \begin{bmatrix} \mathbf{1}_N \, \mathbf{X} \end{bmatrix}^T$ does not affect the commutativity of $\mathbf{X}^T\mathbf{X}$ and $\mathbf{Y}^T\mathbf{Y}$ for the hierarchical dataset, as the constant mode $\mathbf{1}_N$ (i.e., a vector of 1s) is already an eigenvector to both these similarity matrices (see Appendix *Data on a graph* $\mathbf{x} \in \mathbb{R}^{N_{in}}$). Consequently, the analytical solutions by Saxe et al. (2019) remain applicable. We generalize these considerations in a later section on generic input correlations and in the (Appendix *Constant data mode is related to symmetry in the data*).

Fig. 2 shows that linear networks with bias terms have a distinctly different early learning phase when compared to vanilla linear networks. While both models converge to a zero loss solution, we observe that the final network function with bias terms contains an additional non-zero singular value with the associated SVD mode. We devote the next section to analyzing this change in the early dynamics.

## Bias terms drive early learning towards the optimal constant solution

In this section, we qualitatively characterize what causes observed changes in early learning dynamics. We find that early learning dynamics are driven by average output statistics and provide a theoretical explanation. We then demonstrate the generality of this result by highlighting how early learning of average output statistics can be similarly observed in complex, non-linear architectures.

A naive strategy to learning is to minimize error over a set of samples while disregarding information conveyed by the input. Previous work has recently termed this network function the optimal constant solution (OCS) (Kang et al., 2024). The OCS can be formalized as $\hat{\mathbf{y}}_{ocs} = \operatorname{argmin}_{\hat{\mathbf{y}} \in \mathbb{R}^{N_{out}}} \frac{1}{N} \sum_{i=1}^{N} \mathcal{L}(\hat{\mathbf{y}}, \mathbf{y}_i)$ and represents the optimal function $\hat{\mathbf{y}}$ that is independent of input $\mathbf{x}_i$. For mean-squared error, it is straightforward to show that the minimizer is the average output $\hat{\mathbf{y}}_{ocs} = \frac{1}{N} \sum_{i}^{N} \mathbf{y}_i =: \bar{\mathbf{y}}$.

**Setup.** We train linear networks and Convolutional neural networks (CNN) on the hierarchical learning task illustrated in Fig. 1C and D respectively. For CNNs we design a "hierarchical MNIST" task whereby one-hot inputs are replaced with eight randomly sampled classes from MNIST (Li Deng, 2012). For the "hierarchical MNIST" task we use the standard ten image classes provided by MNIST and then sample 8 classes randomly. For each class we then replaced the default one-hot label corresponding to each class $i$ with the corresponding hierarchical, "three-hot" label $\mathbf{y}_i$ seen in Fig. 1B. We use standard uniform Xavier initialization (Glorot & Bengio, 2010) and trained CNNs on an squared error loss. A full description of the CNN experiment and hyperparameter settings is deferred to the Appendix *CNN datasets and hyperparameters*. We also replicate our results with CIFAR-10 (Krizhevsky, 2009), non-hierarchical tasks, alternative loss functions, and CelebA

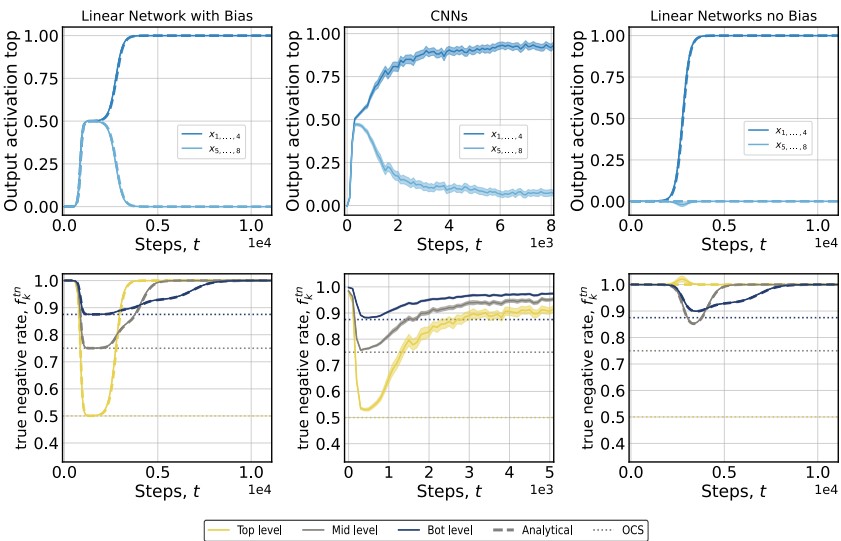

Figure 3: Early learning is driven to the OCS. *Top row:* Network predictions for a single output unit associated with the top level of of the hierarchy in response to all inputs $\mathbf{x}_i$ We see clearly how CNNs and linear networks *with bias* initially change responses while not differentiating between different inputs before learning the correct input output mapping. *Bottom row:* True negative rates, $f_k^{tn}$ for the three hierarchical levels as indicated by colors. For CNNs and linear networks *with bias* Performance approaches levels expected under the OCS (dotted lines).

(Z. Liu et al., 2015) in Appendix *Additional experimental results*. We also show results for shallow linear networks in Appendix *Shallow network OCS learning*.

To assess OCS learning we calculate true negative rates $f_k^{tn}(\mathbf{y}, \hat{\mathbf{y}}) = \frac{(\mathbf{1}_N - \hat{\mathbf{y}}_k)^T(\mathbf{1}_N - \mathbf{y}_k)}{(\mathbf{1}_N - \mathbf{y}_k)^T(\mathbf{1}_N - \mathbf{y}_k)}$ for our task where the subscript $k$ selects the vector slice corresponding to level $k$ of the output hierarchy. We calculate the metric separately for the three hierarchical levels. Effectively, the metric describes how strongly model predictions $\hat{\mathbf{y}}$ align with the desired outputs $\mathbf{y}$ while focusing on zero entries only. The use of the metric is motivated by our desire to highlight how OCS learning is dependent on the distribution of labels in $\mathbf{Y}$. The metric effectively measures wrong beliefs about the presence of target labels across the different levels of the hierarchy. Furthermore, the metric enables later comparisons to human learners (further details in Appendix *True Negative Rates*).

**Empirical evidence**

We identify three separate empirical observations that support early learning of the OCS:

**Indifference.** Linear networks and CNNs initially change outputs while not differentiating between input examples. In Fig. 3 (top) we show the empirical and analytical activation of an output unit associated with the highest level of the hierarchy for all $\mathbf{x}_i$. Networks with and without bias terms learn to differentiate inputs correctly. However, networks with bias terms produce input-independent, non-zero outputs in early training as would be expected under the OCS.

**Performance.** Networks with bias terms show an initial tendency to over-select labels associated with the top level of the hierarchy as seen in the true negative rate $f_k^{tn}(\mathbf{y}, \hat{\mathbf{y}})$ in Fig. 3 (bottom). Furthermore, linear networks and CNNs with bias terms almost exactly approach performance levels that would be provided by the OCS (dotted lines) for each of the three hierarchical levels. Linear network without bias terms do not produce this behavior.

**OCS alignment.** The distance between outputs $\hat{\mathbf{y}}_i$ of linear and non-linear networks and $\mathbf{y}_{ocs}$ approaches zero in early training. Fig. 5 (top) shows how the $L_1$ distance of sample-averaged network outputs and the OCS approaches zero before later converging to the desired network function.

**Theoretical explanation**

In this section, we extend the linear network formalism to understand the mechanism behind early learning of the OCS. We first show how bias terms in the input layer are directly related to the OCS. Afterwards, we prove that the OCS is necessarily learned first in these settings.

**The OCS is linked to shared properties.** Having established the applicability of the linear network theory (see Section *Exact learning dynamics with bias terms*) we now seek to understand how the early bias towards the OCS emerges. To this end, notice how bias terms can be written in terms of the constant eigenmode $\mathbf{1}_N$:

**Proposition 2** (The OCS is linked to shared properties)**.** *If $\mathbf{1}_N$ is an eigenvector to the similarity matrix $\mathbf{X}^T\mathbf{X} \in \mathbb{R}^{N \times N}$, then the sample-average $\bar{\mathbf{x}} = \frac{1}{N}\sum_{i=1}^N \mathbf{x}_i$ will be an eigenvector to*

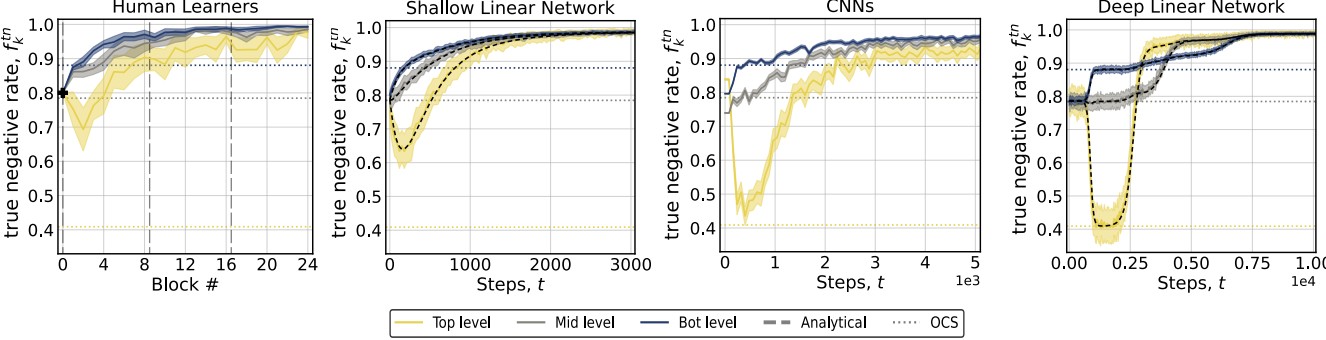

Figure 4: Early response bias towards the OCS across learners in the hierarchical learning task. True negative rates, $f_k^{tn}$ for the three hierarchical levels as indicated by colors for biological and artificial learners (with bias terms). Dotted lines represent performances expected under the OCS. Observe that all learners show a transient bias towards the OCS. Dashed vertical gray lines indicates breaks between days for human learners.

the correlation matrix $\mathbf{X}\mathbf{X}^T \in \mathbb{R}^{N_{in} \times N_{in}}$ with identical eigenvalue $\lambda$. An analogous statement applies for $\mathbf{Y}^T\mathbf{Y}$ and $\mathbf{Y}\mathbf{Y}^T$. The converse does not hold true in general.

We prove this statement in the Appendix (*OCS and shared properties correspond to each other*). Importantly, it establishes a connection between the feature and sample dimensions of $\mathbf{X}$ and $\mathbf{Y}$. If $\mathbf{1}_N$ is an eigenvector to $\mathbf{X}^T\mathbf{X}$ and $\mathbf{Y}^T\mathbf{Y}$ already, it implies that the addition of a bias term will directly add to its eigenvalue, $s_{ocs}^2 \to s_{ocs}^2 + 1$, even if it is initialized at zero. We show in the Appendix (*Data on a graph* $\mathbf{x} \in \mathbb{R}^{N_{in}}$) that these assumptions on $\mathbf{X}$ and $\mathbf{Y}$ hold strictly for our hierarchical task design, and more generally relate to symmetry in the data (*Constant data mode is related to symmetry in the data*). The final section of this paper will discuss how this property extends to natural datasets where exact symmetry is absent.

Crucially, it now follows from Proposition 2 that the time-dependent network correlation $\hat{\mathbf{\Sigma}}^{yx}(t) = \mathbf{U}\mathbf{A}(t)\mathbf{V}^T$ in Eq. (2) will contain a strongly amplified OCS mode $a_{ocs}(t)\mathbf{u}_{ocs}\mathbf{v}_{ocs}^T = a_{ocs}(t)\bar{\mathbf{y}}\bar{\mathbf{x}}^T$ by virtue of the modified singular value $\sqrt{s_{ocs}^2 + 1}$ entering Eq. (2) and thereby the network function. Consequently, learning dynamics will be driven by the outer product of average input and output data. Moreover, this implies that given some input $\mathbf{x}_i$ to Eq. (2), the network's OCS mode contributes

$$\hat{\mathbf{y}}_{ocs}(\mathbf{x}_i) = a_{ocs}(t)\mathbf{u}_{ocs}\mathbf{v}_{ocs}^T\mathbf{x}_i = a_{ocs}(t)\bar{\mathbf{y}}\bar{\mathbf{x}}^T\mathbf{x}_i \propto \bar{\mathbf{y}}. \quad (5)$$

The OCS mode in the time-dependent network function will hence necessarily drive responses towards average output statistics. Note that Eq. (5) also highlights that the more an input example is aligned to average inputs, the more the network's responses will reflect average outputs. In particular, this makes the expected output $\mathbb{E}_{\mathbf{x}}[\hat{\mathbf{y}}(\mathbf{x})] \propto \bar{\mathbf{y}}$. Throughout learning, the evolution of $a_{ocs}(t)$ and scale-dependent alignment of $\mathbf{x}_i$ and $\bar{\mathbf{x}}$ will determine the network's reliance on the OCS mode.

**Early learning is biased by the OCS mode.** We established that network responses are driven by average output statistics $\bar{\mathbf{x}}$ and $\bar{\mathbf{y}}$, but why are *early* dynamics in particular influenced by

the OCS? The learning speed of the SVD modes in the time-dependent network function are controlled by the magnitude of singular values $s_\alpha$ as seen in Eq. (3).

**Theorem 1** (Early learning is biased by the OCS mode). *If* $\mathbf{1}_N$ *is a joint non-degenerate eigenvector to elementwise-positive input and output similarity matrices* $\mathbf{X}^T\mathbf{X}$ *and* $\mathbf{Y}^T\mathbf{Y}$, *the OCS mode* $s_{ocs}\bar{\mathbf{y}}\bar{\mathbf{x}}^T$ *will have leading spectral weight* $s_0 \equiv s_{ocs}$ *in the SVD of the input-output correlation matrix* $\mathbf{\Sigma}^{yx}$.

We prove this statement with help of the Perron-Frobenius theorem (Perron, 1907) in the Appendix (*Constant data mode is the leading eigenvector*). Consequently, the optimal constant mode is learned at a faster rate than remaining SVD components and transiently dominates the early network function. Notably, this applies to our task data $\mathbf{Y}^T\mathbf{Y}$ (see Appendix *Data on a graph* $\mathbf{x} \in \mathbb{R}^{N_{in}}$) and leads to characteristic learning signatures observed in Fig. 3.

Theorem 1 hinges on the constant eigenvector $\mathbf{1}_N$ being present in the data. We later provide empirical (see Fig. 6 and Appendix *Additional experimental results*) and theoretical (Appendix *Constant data mode is related to symmetry in the data*) arguments that this assumption is approximately fulfilled in a variety of cases.

To recapitulate this section: We first rephrased a learnable bias term in the architecture as a shared feature in the input data. We then found that the associated singular value in Eq. (2) drives the learned network function towards the OCS (Eq. (5)). Finally, we proved for linear networks that the bias affects *early* learning. In Appendix *Integrated formulation of architectural biases*, we additionally summarize these results through the neural tangent kernel and highlight that exponential OCS learning can also be induced by bias terms on the output layer of the network. Overall, these results demonstrate that bias terms induce early OCS learning.

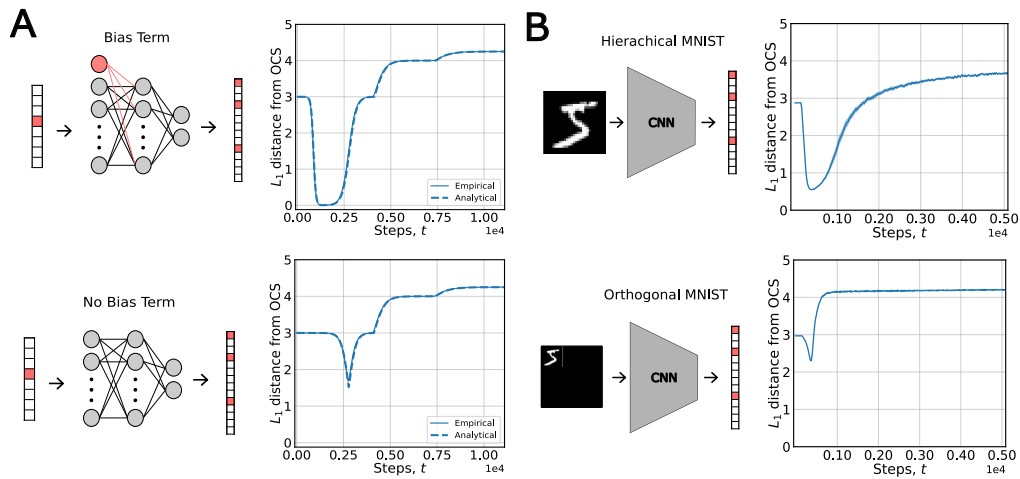

Figure 5: $L_1$ Distance from the OCS in Linear networks and CNNS. **A** Linear networks with (top) and without bias terms (bottom) trained on the hierarchical task **B** CNNs without bias terms trained on variants of the hierarchical task. *Top*: Normal inputs. *Bottom*: "orthogonal" image inputs which remove all between-class input correlations from the input data. Note, how CNNs do not learn the OCS in the absence of input correlations and bias terms.

## OCS learning in humans linear networks, and complex models

In this section, we demonstrate how human learners, linear networks, and non-linear architectures show strong similarities in their early learning on the hierarchical task displayed in Fig. 1.

**Setup.** The hierarchical learning task has previously been used extensively in the study of semantic cognition (Rogers & Mc-Clelland, 2004) and requires learners to develop a hierarchical one-to-many mapping as seen in Fig. 1B. We adapted the task for human learners ($N = 10$) while maintaining the underlying structure: Input stimuli were represented as different classes of planets and output labels were represented as a set of plant images (see Fig. 1E and Fig. 7). We also trained CNNs as in the previous section. Importantly, the hierarchical structure results in a non-uniform distribution of labels with average labels equal to $\mathbf{y}_{ocs}$. Human learners received training over three days. On each trial, participants saw an input (planet) and selected a subset of labels (plants). Subjects received fully informative feedback. A full description of the experimental paradigm is given in the Appendix (*Human learning experiment*). We then compute true negative rate $f_k^{tn}(\mathbf{y}, \hat{\mathbf{y}})$ as in Section *Bias terms drive early learning towards the optimal constant solution* while splitting performance across the hierarchical levels.

Neural networks produced continuous outputs in $\mathbb{R}^{N_{out}}$ while humans responded via discrete button clicks in $\{0,1\}^{N_{out}}$. As we do not have access to "human logits" before response execution we discretized network responses to enable comparison. We treat neural network responses in $\hat{\mathbf{y}}$ as logits from which we then sampled responses in $\{0,1\}^{N_{out}}$. Full procedure details are given in Appendix *Discretizing network responses for comparison to humans*.

**Results.** The key results of our experiments are presented in Fig. 4. Intriguingly, we find that human learners, linear

networks, and CNNs all display characteristic early response biases. Note that chance true negative rate is equal between all three levels of the hierarchy. Biological as well as artificial learners display an initial "drop" in true negative rate at the top level of the output hierarchy. The result indicates a general lack of specificity and an overly liberal response criterion for output labels on the top level of the hierarchy. To appreciate the significance of this result it is important to understand that the task can be learned without the development of these early response biases: In particular, linear networks without bias terms do not show this behavior (see Appendix *true negative rates in linear networks without bias terms*). Surprisingly, the human response signature demonstrates that these learners, just as artifical networks, display an early bias towards the OCS. We conjecture that early learning of the OCS might be a general phenomenon that emerges during error-corrective training. We replicate the human result with a second cohort of learners (see Appendix *Human learning experiment*).

Notable is also the difference between shallow and deep linear networks. Response biases seem more transient in shallow networks and appear to more closely mirror human learners. However, quantitative comparisons are challenging due to inherently differing learning timescales.

## Generic input correlations can equivalently drive OCS learning

We have established how the earliest phase of learning in linear networks is driven by the OCS. In linear networks OCS learning can be induced by bias terms in the network architecture. However, in non-linear architectures, such as CNNs, the network is driven towards the OCS even in the absence of bias terms (Fig. 5B, Top). The appearance of the data term $\mathbf{X}^T\mathbf{X}$ in Proposition 3 suggests an equivalent effect that is induced by the data itself.

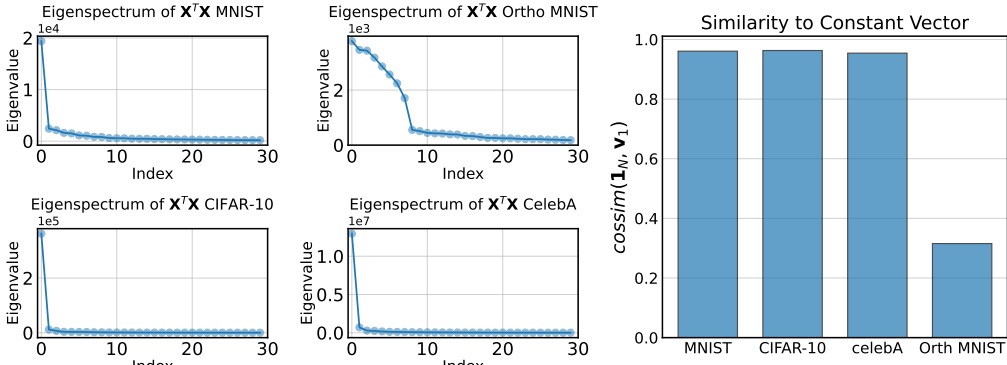

Figure 6: Dataset eigenspectra and constancy of first eigenvector for different image datasets. *Left:* Eigenspectra of $\mathbf{X}^T\mathbf{X}$ for different datasets. *Right:* Alignment of first eigenvector in $\mathbf{X}^T\mathbf{X}$ with the constant vector $\mathbf{1}_N$.

**Corollary 1** (Input correlations induce early OCS)**.** *If $\mathbf{1}_N$ is an eigenvector of the data similarity matrix $\mathbf{X}^T\mathbf{X}$ with non-degenerate eigenvalue $s_0$, then the OCS response during early learning will be driven according to its magnitude.*

This statement follows directly from the joint diagonalisation of Eq. (1) and subsequent projection onto the OCS $\mathbf{y}_{ocs}$. We show a solvable case of OCS learning in linear networks under input correlations and in the absence of bias terms in the Appendix under *OCS-learning in linear networks with input correlations*. We furthermore hypothesize that neural networks will be driven towards the OCS if training data contains more generic input correlations where $\mathbf{1}_N$ is not an exact eigenvector.

**Setup.** We trained CNNs on the hierarchical task (see Section *Bias terms drive early learning towards the optimal constant solution*). Inputs were given by eight randomly sampled classes of MNIST (Fig. 5B, Top). To isolate the effect of input correlations we created a second dataset where randomly sampled classes of MNIST were copied on orthogonal subspaces of a larger image (Fig. 5B, bottom). Importantly, this procedure removes all between-class correlations.

**Results.** The main result of our experiment is displayed in Fig. 5. CNNs which learn from standard MNIST images are strongly driven towards the OCS. In contrast, early dynamics for the "orthogonal" MNIST do not display this tendency. Strikingly, the early dynamics with standard MNIST classes are highly similar to those observed in linear network with bias terms, while the dynamics for the latter task resemble those seen in the linear network without this feature. To verify that generic input correlations are indeed causing these differences we explore the eigenspectrum of the data correlation matrices. We sample 100 images from all 10 classes and compute a correlation matrix $\mathbf{X}^T\mathbf{X}$ from flattened images. First, we find that the eigenspectrum for standard MNIST images is dominated by a single eigenvector (Fig. 6, top-left). In contrast, the eigenspectrum of the orthogonal MNIST task does not display this property (Fig. 6, top-center). Further, recall that input bias terms lead to a non-degenerate constant eigenvector $\mathbf{1}_N$ in the input correlation matrix. Similarly, we find that the first eigenvector $\mathbf{v}_1$ of $\mathbf{X}^T\mathbf{X}$ is indeed highly aligned to $\mathbf{1}_N$ (Fig. 6,

right), whereas this is not the case in the orthogonal MNIST. We additionally show similar results for CIFAR-10 and CelebA. Theoretical considerations suggest that these correlations originate from an approximate symmetry in the data (Appendix *Constant data mode is related to symmetry in the data*).

Overall, we here demonstrated that early learning of the OCS can be driven by properties of the architecture (bias terms) or data (input correlations). Our results also highlight that input correlations are a common feature of standard image datasets: Early learning of the OCS might be a common occurrence when learning from such data.

## Discussion

In this work, we found that the inclusion of bias terms in linear networks shifts early learning towards the OCS, even when initialized at zero. We also highlight how OCS learning can equivalently be driven by input correlations. We demonstrated that early, input-independent OCS biases affect both human learners and non-linear networks. Our contribution complements prior work on early simplicity biases by highlighting factors that drive networks towards early OCS learning; connecting input biases, output statistics, and architecture. Overall, our findings highlight how simple linear networks can serve as useful tools to investigate simplicity biases in more complex systems.

**Relevance.** We see promising applications for early OCS learning in the cognitive and behavioral sciences. OCS-like response biases have been noted previously (Herrnstein, 1961; Estes, 1964). However, we believe that a normative theory for these effects is still incomplete. Our theory identifies possible properties of the biological wetware or natural stimuli that may give rise to such biases.

While we do not study generalization ourselves, we believe that OCS learning is practically important to understand *how* neural networks generalize or fail to generalize. Kang et al. (2024) has highlighted that networks will revert to OCS in a variety of generalization settings. We demonstrate that the OCS component in the network function is acquired *early*, and is *retained* throughout training (effective singular values in Fig. 2 stay constant in late training). We believe that this

retention of the OCS mode enables reversion.

OCS learning is also relevant when learning under class imbalance, a common problem in machine learning where datasets are frequently naturally imbalanced (V. Feldman, 2020; Van Horn & Perona, 2017), leading to a failure to learn about minority classes (Ye et al., 2021). In Appendix *Linear networks under class imbalance* we show a solvable case of OCS learning in such settings and highlight how OCS learning can negatively impact performance on minority classes.

**Limitations and future work.** Our work is restricted to qualitative comparisons between linear networks and non-linear systems and only gives suggestive evidence of factors which drive early OCS learning in non-linear systems. We chose linear networks to allow for a rigorous description of the dynamics of learning. Methods from mean-field theory may provide a precise tool to analyze a wider range of systems directly. Second, the ambiguity between architecture and data in driving the OCS does not allow us to determine the underlying mechanism in human learners. Future studies might address this limitation by manipulating correlations in stimuli or via neural data.

## Acknowledgments

We thank Satwik Bhattamishra, Aaditya Singh, Clémentine Dominé, Lukas Braun, Devon Jarvis, and Kevin Huang for useful feedback, discussions, and comments on earlier versions of this work. This work was funded by a European Research Council (ERC) Consolidator Award (725937) to C.S., a Wellcome Trust Discovery Award (227928/Z/23/Z) to C.S., and a UKRI ESRC Grand Union Doctoral training partnership stipend awarded to J.R. This work was also supported by a Schmidt Science Polymath Award to A.S., and the Sainsbury Wellcome Centre Core Grant from Wellcome (219627/Z/19/Z) and the Gatsby Charitable Foundation (GAT3850). A.S. is a CIFAR Azrieli Global Scholar in the Learning in Machines & Brains program.

## Author Contributions

The conceptualisation of the project was developed by J.R. with C.S. and A.S. providing supervision. J.R. is responsible for all empirical results in networks, the human experiment, and initial theoretical ideas. J.B. and J.R. collaboratively devised the main theoretical results in the paper. J.B. primarily developed the theoretical presentation in the appendix. The initial draft was written by J.R. with further iterations and appendices written collaboratively by J.B. and J.R. All authors contributed to polishing of the draft and C.S. and A.S. provided supervision on all aspects of the project.

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

# Appendix / supplemental material

## Overview

Our appendix has the following sections:

- In Appendix *Human learning experiment*, we describe the human experiment in more detail and show the results of a replication in a second cohort. We furthermore report statistical tests and describe ethical considerations.

- In Appendix *Discretizing network responses for comparison to humans*, we outline how we bring neural network and human responses displayed in Section *OCS learning in humans linear networks, and complex models* into a common space for direct comparison.

- In Appendix *True Negative Rates*, we outline how we compute the true negative rates, $f^{tn}$ used in Section *Bias terms drive early learning towards the optimal constant solution* and Section *OCS learning in humans linear networks, and complex models*.

- In Appendix *Additional theoretical results and proofs*, we provide additional theoretical derivations and remaining proofs to the statements in the main text.

- In Appendix *Shallow network OCS learning*, we show OCS signatures in *shallow* networks with bias terms.

- In the short Appendix *true negative rates in linear networks without bias terms*, we show how linear networks without bias terms behave on the task in Section *OCS learning in humans linear networks, and complex models*.

- In Appendix *CNN datasets and hyperparameters*, we describe hyperparameters, datasets, and further training details used for our CNN experiments.

- In Appendix *Additional experimental results*, we describe the results of additional experiments investigating early emergence of the OCS in non-linear models.

- In Appendix *Linear networks under class imbalance* we show an additional solvable case of linear networks with bias terms under class imbalance.

- In Appendix *OCS-learning in linear networks with input correlations* we show OCS learning in linear networks with input correlations but in the absence of bias terms.

- In Appendix *Relation to imbalanced multi-label learning* we discuss additional connections of our work to multi-label learning.

## Human learning experiment

We directly translated the hierarchical task setup used by Saxe et al. (2019) into an experimental paradigm. Our design attempts to stay as close to the original task structure used for neural networks as possible. We designed the task as a mapping from 8 distinct input stimuli represented as planets to a set of 3 associated output stimuli represented as plants (see Fig. 7, left).

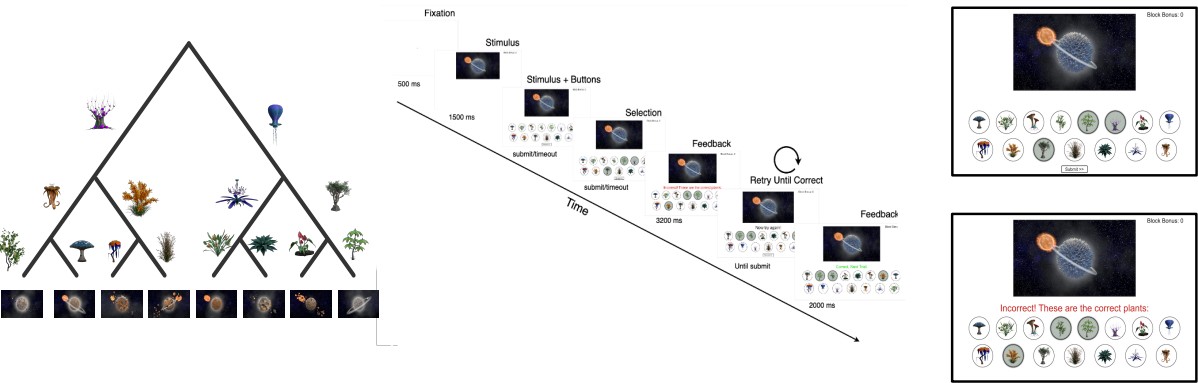

Figure 7: Human task design. *Left:* Hierarchical learning task, adapted for human participants. *Centre:* Trial structure as experienced by human participants. *Right:* Example screen during response period (top), Example screen during feedback period (bottom).

In the task, participants had to learn to associate which outputs properties are associated with each input. Unbeknownst to the participants we imposed a hierarchical structure on output targets (Fig. 7, left). In the structure some output labels are associated with more than one input. As a control for analyses we also included an additional control input-output pair (similarly represented by a planet and a plant; not shown here and excluded from current analysis). We recruited a cohort of 10 subjects that were trained over the course of three days with one daily session. The cohort was recruited as part of a larger neuroimaging experiment but our analysis presented here is exclusive to behavioural results. We further replicated our results in a second cohort of 46 human subjects recruited via the online platform Prolific (prolific.com). Results of the replication of the study can be seen in Fig. 8.

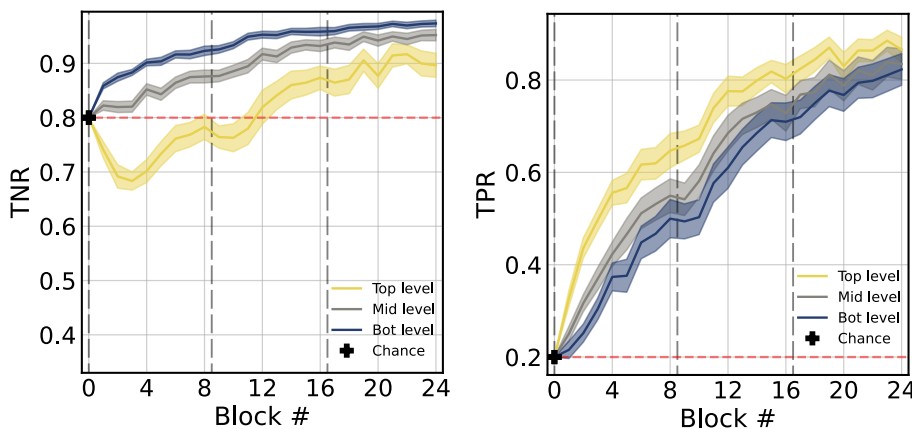

Figure 8: The human replication cohort. While learning is slower, the qualitative pattern indicating reliance on the OCS is replicated. *Left*: TNR rate. *Right:* TPR rate.

Each day of at home training consisted of 8 blocks of training with 22 trials each (160 standard trials and 16 control trials) which lasted about one hour. The trial structure during training is shown in Fig. 7, centre. During training trials, subjects were shown the stimulus on screen and were required to press three buttons, presented below the planet image (Fig. 7, right). The subjects received fully informative feedback on each trial and were forced to repeat the trial in the case of incorrect selection until the correct properties were selected. The location of buttons was shuffled on screen for each trial and for each forced repetition. For each button clicked correctly on their first attempt participants received a bonus point. We displayed a block-wise bonus in the corner of the screen throughout the task. Participants were payed slightly above local minimum wage as a baseline and received a substantial performance dependent bonus (on average about one-third of the baseline pay). We include a screenshot of the initial instructions in Fig. 9. Beyond this initial instruction screen participants received more nuanced instructions about clicking of buttons and feedback in the beginning of the task.

**Statistical tests.** While our focus is on qualitative patterns in human behaviour, we compute statistical tests on the true negative rates for human results seen in the main text (Fig. 4). We averaged all blocks in a given day and performed a two-way repeated measures ANOVA to assess the effect of day and hierarchy level on true negative rates. The two-way repeated measures ANOVA revealed significant main effects of day $F(2, 18) = 57.22$, $p < .0001$, $\eta^2 = .25$ and level $F(2, 18) = 6.25$, $p = .033$, $\eta^2 = .18$. Beyond this we also found a significant interaction of day and level $F(4, 36) = 9.795$, $p = .0056$, $\eta^2 = .042$. A Mauchly test indicated that the assumption of sphericity had been violated for level $\chi^2 = .03$, $p < .5$ and the interaction term $\chi^2 = .006$, $p < .5$. Significance values are reported with Greenhouse-Geisser correction. The results confirm that performance between levels are significantly different depending on day and hierarchical level.

**Ethical considerations.** Human participants performed a simple, computerised learning task without the collection of personal identifiable information or substantial deception. Human data collection was handled strictly in line with institutional guidelines and under institutional review board approval. We obtained informed consent for each participant before commencing the study. We highlighted that participants could withdraw at any time without penalty or loss of compensation by simply exiting full-screen or informing the experimenter. We provided contact emails in the case of concern or questions. Data was handled in a strictly anonymised format and stored on password secured devices. Participants were payed above minimum wage for their country of origin.

## Discretizing network responses for comparison to humans

**Discretization.** In our task, neural networks produced continuous outputs. This is distinct from human learners who were required to give discrete responses. We now describe the discretization that allows us to compare human and network responses. Fundamentally, we conceptualise inference as a noisy process by which responses are sampled from a distribution over output

This experiment should take about **80 min**. This task requires a lot of attention. Please make sure you are undisturbed for the next 80 minutes.

Figure 9: Initial instructions received by participants after the collection of informed consent.

labels. That is, we treat outputs from our linear network as logits. We first feed network outputs $\hat{\mathbf{y}}_i$ through a softmax function with temperature 0.2 and subsequently sample three responses without replacement. The procedure maps continuous outputs $\hat{\mathbf{y}}_i$ to binary responses vectors in $\{0,1\}^{N_{out}}$.

**Expected solutions.** Here we describe the derivation of expected solutions used in Fig. 4, dashed lines. The derivation of these "expected responses" under the sampling procedure allows to make the reliance of network responses on the exact solutions in Section *Exact learning dynamics with bias terms* clear.

Consider network outputs $\hat{\mathbf{y}}_i(t) = \mathbf{W}^2(t)\mathbf{W}^1(t)\mathbf{x}_i$. We transform these outputs through a softmax function $\sigma_\beta : \mathbb{R}^{N_{out}} \to (0,1)^{N_{out}}$. Let $S = \{s_1, s_2, s_3\}$ denote the set of three unique response indices sampled from $\sigma_\beta(\hat{\mathbf{y}}_i(t))$ without replacement, where $s_n \in \{1,2,\ldots,N_{out}\}$ for $n = 1,2,3$, and all $s_n$ are hence distinct. The probability distribution $\sigma_\beta(\hat{\mathbf{y}}_i(t))$ is dependent on time $t$, therefore denote the produced probability of $S$ as $P_t(S)$. For each of these sets $S$ we can compute an associated true negative rate for each of the $k \in \{1,2,3\}$ levels in the hierarchy. We denote this random variable as $X_S^k$. We can then compute expected solutions to inference behaviour as

$$\mathbb{E}_t[X_S^k] = \sum_{S \subseteq \{1,2,\ldots,m\}, |S|=3} P_t(S)X_S^k \tag{6}$$

### True Negative Rates

Here we describe the metric used in the bottom panel of Fig. 3 and in Fig. 4. The metrics effectively describes true negative rates (correct-rejection scores). We use the metric on continuous network responses in $\mathbb{R}^{N_{out}}$ in Fig. 3. We also use the metric on discretised networks responses in $\{0,1\}^{N_{out}}$ and for human responses in $\{0,1\}^{N_{out}}$ in Fig. 4.

Given responses $\hat{\mathbf{y}}$ and target vectors $\mathbf{y} \in \mathbb{R}^{N_{out}}$ the metric computes the alignment between target and response vectors while only focusing on zero entries in $\mathbf{y}$. Furthermore we compute the metric separately for the $k \in \{1,2,3\}$ separate levels of the hierarchy where the entries $s_k$ and $e_k$ denote relevant start and end indices of level $k$ in the vectors $\hat{\mathbf{y}}$ and $\mathbf{y}$. The metric is then computed as

$$f_k^{tn}(\hat{\mathbf{y}}, \mathbf{y}) = \frac{(\mathbf{1}_{N_{out}} - \hat{\mathbf{y}})_{s_k:e_k}^T (\mathbf{1}_{N_{out}} - \mathbf{y})_{s_k:e_k}}{(\mathbf{1}_{N_{out}} - \mathbf{y})_{s_k:e_k}(\mathbf{1}_{N_{out}} - \mathbf{y})_{s_k:e_k}}, \tag{7}$$

where $s_k : e_k$ is a "slicing" notation that takes the subvector between indices $s_k$ and $e_k$.

If for all desired entries of $0$ in $\mathbf{y}$ the vector $\hat{\mathbf{y}}$ is equal to $0$ the metric will be at 1. Correspondingly if entries in $\hat{\mathbf{y}}$ are larger than zero the metric $f_k^{tn}(\hat{\mathbf{y}}, \mathbf{y})$ will decrease. Thus, the metric measures wrong beliefs about the presence of target labels across the different levels of the hierarchy.

### Additional theoretical results and proofs

**Equivalence of bias terms**   In this section, we give more detail on the method used in in Section *Exact learning dynamics with bias terms* of how to reformulate a bias term in terms of the network weights and a constant feature in the input.

Consider a network with an explicit input bias term $\mathbf{b}^1$,

$$\hat{\mathbf{y}} = \tilde{\mathbf{W}}^1\tilde{\mathbf{x}} + \tilde{\mathbf{b}}^1$$

This is equivalent to introducing a constant component to the vector $\mathbf{x}$,

$$\tilde{\mathbf{x}} \to \mathbf{x} := \begin{bmatrix} 1 \\ \tilde{\mathbf{x}} \end{bmatrix},$$

and using the network

$$\hat{\mathbf{y}} = \mathbf{W}^1\mathbf{x},$$

*Proceedings of Cognitive Computational Neuroscience 2025*

as we can write

$$\left(\mathbf{W^1 x}\right)_m = \sum_{j=0}^{N_{in}} W_{mj}^1 x_j$$

$$= W_{m0}^1 1 + \sum_{j=1}^{N_{in}} W_{mj}^1 x_j$$

$$= W_{m0}^1 1 + \sum_{j=0}^{N_{in}-1} \tilde{W}_{mj}^1 \tilde{x}_j$$

$$\equiv b_m^1 + \sum_{j=0}^{N_{in}-1} \tilde{W}_{mj}^1 \tilde{x}_j.$$

In order to match a given i.i.d. initialization $b_m^1 \sim \mathcal{N}\left(0, \sigma_b^2\right)$ where $\sigma_b \neq \sigma_w$, the component that needs to be added to $\tilde{\mathbf{x}}$ to get equivalence needs to be $\sigma_b/\sigma_w$.

**Learning dynamics for bias terms** We here derive analytical expressions for the learning speeds of input and output bias terms for a two-layer deep linear network discussed in the main text,

$$\hat{\mathbf{y}} = \mathbf{W}^2\left(\mathbf{W}^1\mathbf{x} + \mathbf{b}^1\right) + \mathbf{b}^2.$$

We decompose $\mathbf{W}^2 = \mathbf{U}\mathbf{A}^{(2)}\mathbf{R}^{(2)}$ and $\mathbf{W}^1 = \mathbf{R}^{(1)}\mathbf{A}^{(1)}\mathbf{V}$ by means of a singular value decomposition (SVD). We here make the assumption of balancedness $\mathbf{W}^1(0)\mathbf{W}^{1T}(0) = \mathbf{W}^{2T}(0)\mathbf{W}^2(0)$ (Braun et al., 2022) at the beginning of training, which implies $\mathbf{R}^{(2)}\mathbf{S}^{(2)2}\mathbf{R}^{(2)T} = \mathbf{R}^{(1)}\mathbf{S}^{(1)2}\mathbf{R}^{(1)T}$. For clarity, we further assume the simplification

$$\mathbf{R}^{(2)T} = \mathbf{R}^{(1)} =: \mathbf{R}, \ \mathbf{A}^{(2)} = \mathbf{A}^{(1)} =: \sqrt{\mathbf{A}}.$$

We here just state these relations without further comment to complement the respective derivation for the weights in (Saxe et al., 2014). This decomposition then allows to rewrite the gradients.

**Input bias term**

$$\tau\frac{d}{dt}\mathbf{b}^1 = \nabla_{\mathbf{b}^1}\mathcal{L}$$

$$= (\mathbf{y} - \hat{\mathbf{y}})^T\mathbf{W}^2$$

$$= \left(\mathbf{y} - \left(\mathbf{W}^2\left(\mathbf{W}^1\mathbf{x} + \mathbf{b}^1\right) + \mathbf{b}^2\right)\right)^T\mathbf{W}^2$$

$$\mathbb{E}_{\mathbf{x}} \to \left(\bar{\mathbf{y}} - \mathbf{W}^2\left(\mathbf{W}^1\bar{\mathbf{x}} + \mathbf{b}^1\right) - \mathbf{b}^2\right)^T\mathbf{W}^2$$

$$= \left(\bar{\mathbf{y}} - \mathbf{U}\mathbf{A}\mathbf{V}\bar{\mathbf{x}} - \mathbf{U}\sqrt{\mathbf{A}}\mathbf{R}\mathbf{b}^1 - \mathbf{b}^2\right)^T\mathbf{U}\sqrt{\mathbf{A}}\mathbf{R}^T$$

$$= \bar{\mathbf{y}}^T\mathbf{U}\sqrt{\mathbf{A}}\mathbf{R}^T - \bar{\mathbf{x}}^T\mathbf{V}^T\mathbf{A}\mathbf{R}^T - \mathbf{b}^{1T}\mathbf{R}\mathbf{A}\mathbf{R} - \mathbf{b}^{2T}\mathbf{U}\sqrt{\mathbf{A}}\mathbf{R}^T$$

$$= (\mathbf{Y}\mathbf{1}_N)^T\mathbf{U}\sqrt{\mathbf{A}}\mathbf{R}^T - (\mathbf{X}\mathbf{1}_N)^T\mathbf{V}^T\mathbf{A}\mathbf{R} - \mathbf{b}^{1T}\mathbf{R}\mathbf{A}\mathbf{R}^T - \mathbf{b}^{2T}\mathbf{U}\sqrt{\mathbf{A}}\mathbf{R}^T.$$

Here, we denoted the expectation over the data samples as $\mathbb{E}_{\mathbf{x}}$. Projecting from the right with $\mathbf{R}_\alpha \in \mathbb{R}^{N_{\text{hidden}}}$ gives

$$\tau\frac{d}{dt}\left(\mathbf{b}^{1T}\mathbf{R}_\alpha\right) = \bar{\mathbf{y}}^T\mathbf{U}_\alpha\sqrt{a_\alpha} - \bar{\mathbf{x}}^T\mathbf{V}_\alpha^T a_\alpha - \mathbf{b}^{1T}\mathbf{R}_\alpha a_\alpha - \mathbf{b}^{2T}\mathbf{U}_\alpha\sqrt{a_\alpha}. \tag{8}$$

**Output bias term**

$$\tau\frac{d}{dt}\mathbf{b}^2 = \left(\mathbf{y} - \left(\mathbf{W}^2\left(\mathbf{W}^1\mathbf{x} + \mathbf{b}^1\right) + \mathbf{b}^2\right)\right)$$

$$\mathbb{E}_{\mathbf{x}} \to \bar{\mathbf{y}} - \mathbf{W}^2\left(\mathbf{W}^1\bar{\mathbf{x}} + \mathbf{b}^1\right) - \mathbf{b}^2$$

$$= \bar{\mathbf{y}} - \mathbf{U}\mathbf{A}\mathbf{V}\bar{\mathbf{x}} - \mathbf{U}\sqrt{\mathbf{A}}\mathbf{R}^T\mathbf{b}^1 - \mathbf{b}^2$$

$$= \mathbf{Y}\mathbf{1}_N - \mathbf{U}\mathbf{A}\mathbf{V}\mathbf{X}\mathbf{1}_N - \mathbf{U}\mathbf{R}^T\mathbf{b}^1 - \mathbf{b}^2. \tag{9}$$

Notably, the derivative in Eq. (8) is proportional to the singular vectors of the weights $a_\alpha$, so that its growth is attenuated, analogous to the sigmoidal growth in deep linear networks (Saxe et al., 2014). In contrast, the learning signal $\frac{d}{dt}\mathbf{b}^2$ in Eq. (9) is not affected by the initialization of the weights and is hence $O(1)$ already at the beginning of learning, reminiscent of shallow networks.

**Integrated formulation of architectural biases.** In the main text, we have analysed how bias terms on the *input* layer affect the singular value spectrum. Our empirical results in Section *Empirical evidence* suggest a more general dynamical bias towards the OCS stemming purely from architectural properties. Here, we use the neural tangent kernel $\mathrm{NTK}(\mathbf{x}_i, \mathbf{x}_{i'}) = \sum_k \frac{d\hat{\mathbf{y}}_i}{d\theta_k} \frac{d\hat{\mathbf{y}}_{i'}^T}{d\theta_k}$ (Jacot et al., 2018) to directly and comprehensively describe the affected time evolution of the network response $\frac{d}{dt}\hat{\mathbf{y}}_i = \mathrm{NTK}(\mathbf{y}_i - \hat{\mathbf{y}}_i)$ at the cost of a closed-form solution. Because changes in network outputs are proportional to the NTK it can been viewed as an architecture-induced learning rate (Roberts et al., n.d.). For a review and derivation of the NTK, see Appendix *Neural tangent kernel*. For completeness, we now consider a network that contains input $\mathbf{b}^1$ and output $\mathbf{b}^2$ bias terms.

**Proposition 3** (NTK of linear networks with bias terms). *Consider a two-layer linear network with input and output-layer bias $\hat{\mathbf{Y}} = \mathbf{W}^2(\mathbf{W}^1\mathbf{X} + \mathbf{b}^1) + \mathbf{b}^2$ in the high-dimensional regime. Furthermore, assume weights are initialized i.i.d. $W_{ij}^\ell \sim \mathcal{N}(0, \sigma_{\mathbf{W}^\ell}^2/N_{in}^\ell)$ in each layer. Then, the neural tangent kernel of in early training in expectation $\mathbb{E}_{\mathbf{W}}$ reads*

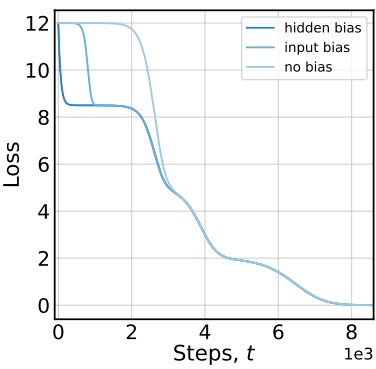

Figure 10: Loss curves for different bias variations.

$$\mathrm{NTK}(\mathbf{X},\mathbf{X}) = \sigma_{\mathbf{W}^2}^2 \mathbf{I}_{N_{out}} \otimes \left(2\mathbf{X}^T\mathbf{X} + \underbrace{\mathbf{1}_N\mathbf{1}_N^T}_{\leftrightarrow\mathbf{b}^1}\right) + \mathbf{1}_{N_{out}}\mathbf{1}_{N_{out}}^T \otimes \underbrace{\mathbf{1}_N\mathbf{1}_N^T}_{\leftrightarrow\mathbf{b}^2}. \tag{10}$$

The tensor product $\otimes$ separates the components that operate on output and sample space. We briefly review the NTK and derive this expression in the next section. The highlighted terms originate from the bias term $\frac{d\hat{\mathbf{y}}_i}{d\mathbf{b}} \frac{d\hat{\mathbf{y}}_{i'}^T}{d\mathbf{b}}$ entering the NTK, manifesting in the appearance of the constant mode $\mathbf{1}_N$. Importantly, these terms do not scale with the size of the learned bias – they are present even if the bias is initialized at zero. Intuitively, their contribution stems from the architecture's *potential* to learn a bias, enabling rapid changes in output $\hat{\mathbf{y}}$. The NTK also reveals a qualitative difference between input and output bias: Whereas the term that is induced by $\mathbf{b}^1$ shows attenuated growth due to the multiplication by the weights of initial scale $\sigma_{\mathbf{W}^2} \ll 1$, the output bias $\mathbf{b}^2$ immediately changes the output significantly. Loss curves which demonstrate the effect of different bias terms are displayed in Fig. 10.

## Proofs

### Feasibility of closed-form solution

**Proposition 1** (Feasibility of closed-form learning dynamics). *For any input data $\mathbf{X} \in \mathbb{R}^{N_{in}\times N}$ and output data $\mathbf{Y} \in \mathbb{R}^{N_{out}\times N}$ it is possible to diagonalize $\boldsymbol{\Sigma}^x$ by the right singular vectors $\mathbf{V}$ of $\boldsymbol{\Sigma}^{yx}$ if $\mathbf{Y}^T\mathbf{Y}$ and $\mathbf{X}^T\mathbf{X}$ commute. The converse holds true only if $\mathbf{X}$ has a left inverse.*

*Proof.* We would like to know when the right singular vectors $\mathbf{V}$ (denote as $\mathbf{V}^{yx}$ here for clarity) of $\boldsymbol{\Sigma}^{yx} = \mathbf{U}^{yx}\mathbf{S}^{yx}\mathbf{V}^{yx}$ match these of $\boldsymbol{\Sigma}^x = \mathbf{U}^x\mathbf{S}^x\mathbf{V}^x$. First, to reduce the problem to $\mathbf{V}^{yx}$, note that $\boldsymbol{\Sigma}^{yxT}\boldsymbol{\Sigma}^{yx} = \mathbf{V}^{yx}\mathbf{S}^{yx2}\mathbf{V}^{yx}$, so that what remains to show is $\left[\boldsymbol{\Sigma}^{yxT}\boldsymbol{\Sigma}^{yx}, \boldsymbol{\Sigma}^{xx}\right] = 0$, where $[\mathbf{A}, \mathbf{B}] := \mathbf{AB} - \mathbf{BA}$ denotes the commutator between two matrices $\mathbf{A}$ and $\mathbf{B}$. We compute the two terms as

$$\boldsymbol{\Sigma}^{yxT}\boldsymbol{\Sigma}^{yx}\boldsymbol{\Sigma}^{xx} = \mathbf{X}\mathbf{Y}^T\mathbf{Y}\mathbf{X}^T\mathbf{X}\mathbf{X}^T$$

$$\boldsymbol{\Sigma}^{xxT}\boldsymbol{\Sigma}^{yxT}\boldsymbol{\Sigma}^{yx} = \mathbf{X}\mathbf{X}^T\mathbf{X}\mathbf{Y}^T\mathbf{Y}\mathbf{X}^T$$

The commutator vanishes if these terms match, which happens for the simpler equality

$$\mathbf{Y}^T\mathbf{Y}\mathbf{X}^T\mathbf{X} = \mathbf{X}^T\mathbf{X}\mathbf{Y}^T\mathbf{Y},$$

or $\left[\mathbf{Y}^T\mathbf{Y}, \mathbf{X}^T\mathbf{X}\right] = 0$. The converse follows only if the transformation $\mathbf{X}\dots\mathbf{X}^T$ in the former equation is invertible, which is the case if a left inverse $\mathbf{X}^{-1}\mathbf{X} = \mathbf{I}_{N_{in}}$ exists.

**OCS and shared properties correspond to each other**   We here link the OCS and shared properties stand in close relation, as the eigenvector $\mathbf{1}_N$ represents properties that are shared across all data samples.

**Proposition 2** (The OCS is linked to shared properties). *If $\mathbf{1}_N$ is an eigenvector to the similarity matrix $\mathbf{X}^T\mathbf{X} \in \mathbb{R}^{N \times N}$, then the sample-average $\bar{\mathbf{x}} = \frac{1}{N}\sum_{i=1}^{N}\mathbf{x}_i$ will be an eigenvector to the correlation matrix $\mathbf{X}\mathbf{X}^T \in \mathbb{R}^{N_{in} \times N_{in}}$ with identical eigenvalue $\lambda$. An analogous statement applies for $\mathbf{Y}^T\mathbf{Y}$ and $\mathbf{Y}\mathbf{Y}^T$. The converse does not hold true in general.*

*Proof.*

$$\mathbf{X}\mathbf{X}^T\bar{\mathbf{x}} = \left(\mathbf{X}\mathbf{X}^T\right)\frac{1}{N}\mathbf{X}\mathbf{1}_N = \frac{1}{N}\mathbf{X}\left(\mathbf{X}^T\mathbf{X}\right)\mathbf{1}_N = \frac{1}{N}\mathbf{X}\lambda\mathbf{1}_N = \lambda\frac{1}{N}\mathbf{X}\mathbf{1}_N = \lambda\bar{\mathbf{x}}.$$

**Constant data mode is related to symmetry in the data**   This section gives proof sketches based on symmetry in the dataset that are sufficient to make $\mathbf{1}_N$ an eigenvector to $\mathbf{X}^T\mathbf{X}$ and $\mathbf{Y}^T\mathbf{Y}$, and in particular hold for the dataset that we are considering. We anticipate that it is possible to formulate these statements in a more universal way by fully leveraging the cited literature.

   The assumptions on symmetry should intuitively at least hold in an approximate manner for many datasets, we expect that they indeed are the reason why we observe a prevalence of $\mathbf{1}_N$, although they are not a necessary condition.

**Continuously supported data $\mathbf{x} \in \mathbb{R}^{N_{in}}$**

**Proposition 4** (Continuous symmetry induces $\mathbf{1}_N$). *If the pairwise correlations $\mathbf{y}_i^T\mathbf{y}_{i'}$ in a dataset are rotationally symmetric, its similarity matrix $\mathbf{Y}^T\mathbf{Y}$ has eigenvector $\mathbf{1}_N$. Note that this is a weaker assumption than the data itself being symmetric.*

*Proof.* We assume that $\mathbf{Y}, \mathbf{X}$ have been sampled from a ground truth data distribution $p(\mathbf{y}, \mathbf{x})$. If $p(\mathbf{y})$ is rotationally symmetric and $\mathbf{X}$ is comprised of samples $\mathbf{x}$ that are uniformly distributed on the hypersphere, we can introduce the kernel function $\mathbf{y}_{\mathbf{x}_i}^T\mathbf{y}_{\mathbf{x}_{i'}} = k(\mathbf{x}_i, \mathbf{x}_{i'}) = k(\mathbf{R}_l\mathbf{x}_i, \mathbf{R}_l\mathbf{x}_{i'}) = k(\mathbf{x}_i^T\mathbf{x}_{i'})$ for any $R_l$ that is a representation of the group of rotations $G = \mathrm{SO}(N_{in})$ that faithfully acts on the "subsampled" hypersphere $\mathbf{X}$ comprised of vectors $\mathbf{x} \in \mathbb{R}^{N_{in}}$. It therefore only depends on the pairwise input similarity (hence sometimes called dot-product kernel). If follows that for all vectors $\mathbf{v}(\mathbf{X}) \in \mathbb{R}^N$ that are evaluations of the functions of the sample points $\mathbf{X}$

$$\mathbf{Y}^T\mathbf{Y}\mathbf{v} = k(\mathbf{X}^T\mathbf{X})\mathbf{v} = k((\mathbf{R}_l\mathbf{X})^T(\mathbf{R}_l\mathbf{X}))\mathbf{v} = \mathbf{R}_k^T k(\mathbf{X}^T\mathbf{X})\mathbf{R}_l\mathbf{v} \Leftrightarrow [\mathbf{Y}^T\mathbf{Y}, \mathbf{R}_l] = 0,$$

where $[\mathbf{A}, \mathbf{B}] =: \mathbf{A}\mathbf{B} - \mathbf{B}\mathbf{A}$ is the commutator between two matrices.
   It follows that we must have for all rotations $\mathbf{R}_l$

$$\mathbf{R}_l\left(\mathbf{Y}^T\mathbf{Y}\mathbf{1}_N\right) = \mathbf{Y}^T\mathbf{Y}\mathbf{R}_l\mathbf{1}_N = \mathbf{Y}^T\mathbf{Y}\lambda_{\mathbf{R}_l}\mathbf{1}_N = \lambda_{\mathbf{R}_l}\mathbf{Y}^T\mathbf{Y}\mathbf{1}_N$$

with eigenvalue $\lambda_{\mathbf{R}_l} = 1$.
   meaning that $\mathbf{Y}^T\mathbf{Y}\mathbf{1}_N$ is an eigenvector to *all* $\mathbf{R}_l$. This can only be the case if $\mathbf{Y}^T\mathbf{Y}\mathbf{1}_N \propto \mathbf{1}_N$, as this is the only vector of values on the sphere that is invariant under any rotations.

   We point out that it can be shown more generally with tools from functional analysis that the full spectrum of this kernel operator $k$ are the spherical harmonics if the data measure $p(\mathbf{x})$ is spherically symmetric (Hecke, 1917), see (Dutordoir et al., 2020) for a modern presentation with tools from calculus. As the first harmonic $\mathcal{Y}_{l=0,m=0}(\mathbf{x})$ is constant, it follows that also the constant function $\mathbf{1}(\mathbf{x}) \equiv 1$ is an eigenfunction when drawing a finite set of samples from this kernel.

**Data on a graph $\mathbf{x} \in \mathbb{R}^{N_{in}}$**   We here prove that the former statement holds for the hierarchical dataset that is discussed in the main text, i.e. that $\mathbf{1}_N$ is an eigenvector to $\mathbf{Y}^T\mathbf{Y}$.

   First, note that it is easy to convince oneself of this by writing down the matrices explicitly: Then, as the rows are just permutations of one another, $\mathbf{1}_N$ is immediately identified as an eigenvector, because $\sum_{i'}^{N}\mathbf{Y}_i^T\mathbf{Y}_{i'}1_{i'} = \mathbf{Y}_i^T\left(\sum_{i'}\mathbf{Y}_{i'}\right)$ will then not depend on $i$ and hence be proportional to $\mathbf{1}_N$.

   To connect with the former symmetry-based argument Appendix *Constant data mode is related to symmetry in the data*, we here however give a proof that is based on the symmetry in the data:

**Proposition 5** (Discrete symmetry induces $\mathbf{1}_N$). *Consider a connected Cayley tree graph with adjacency matrix $\mathbf{A}$ and nodes $\mathbf{x}_i$. Furthermore, let $\mathbf{R}_l \in G$ be an element of a faithful representation of the symmetry group $G$ that acts on the graph nodes $\mathbf{v}$, i.e. that leaves its adjacency matrix invariant, $[\mathbf{R}_l, \mathbf{A}] = 0 \,\forall \mathbf{R}_l$.*

   *If $\mathbf{Y}$ are labels associated with the leaf nodes $\mathbf{X}$ (the outermost generation of the graph, see (Erzan & Tuncer, 2020)) and there exists a similarity function $k$ such that $\mathbf{y}_{\mathbf{x}_i}^T\mathbf{y}_{\mathbf{x}_{i'}} = \mathbf{y}_{\mathbf{R}_l\mathbf{x}_i}^T\mathbf{y}_{\mathbf{R}_l\mathbf{x}_{i'}} = k(d(\mathbf{x}_i, \mathbf{x}_{i'})) \,\forall \mathbf{R}_l$ where $d$ is the geodesic distance on the graph, $\mathbf{1}_N$ will be an eigenvector of $\mathbf{Y}^T\mathbf{Y}$.*

*Proof.* From the symmetry assumption on the labels, we again have for any vector $\mathbf{v}$ of node loadings $\left[\mathbf{R}_l, \mathbf{Y}^T\mathbf{Y}\right]\mathbf{v} = 0 \,\forall\, \mathbf{R}_l \in G$. From this, we find that

$$\mathbf{R}_l\left(\mathbf{Y}^T\mathbf{Y}\mathbf{1}_N\right) = \mathbf{Y}^T\mathbf{Y}\mathbf{R}_l\mathbf{1}_N = \mathbf{Y}^T\mathbf{Y}\lambda_{\mathbf{R}_l}\mathbf{1}_N = \lambda_{\mathbf{R}_l}\mathbf{Y}^T\mathbf{Y}\mathbf{1}_N \,\forall\, \mathbf{R}_l$$

This shows that $\mathbf{Y}^T\mathbf{Y}\mathbf{1}_N$ is an eigenvector of $\mathbf{R}_l$ with eigenvalue $\lambda_{\mathbf{R}_l} = 1$ for any element of the symmetry group. The only vector $\mathbf{v}$ that is invariant under *all* symmetry operations of the graph is the constant vector $\mathbf{1}_N$.

We briefly point out the rich literature on spectral graph theory (for example (Brouwer & Haemers, 2011; Erzan & Tuncer, 2020)) that might allow making statements about the nature of the eigenvalues and other eigenvectors as a function of the graph topology. We expect that this is possible because the literature in the continuous case discussed in the next paragraph bases their arguments on the Laplacian on the sphere, an operator that can be extended to graphs as well. We leave these exploration for future work.

**Corollary 2.** *Because $k(\mathbf{X}^T\mathbf{X}) := \mathbf{X}^T\mathbf{X}$ defines a particular case of input-output similarity mapping, $\mathbf{1}_N$ is also an eigenvector to $\mathbf{X}^T\mathbf{X}$ under the former assumptions of uniform data distribution.*

**Constant data mode is the leading eigenvector** Here, we prove that the constant eigenvector $\mathbf{1}_N$ which is responsible for the OCS solution is associated with the *leading* eigenvalue of the input-output correlation matrix and hence drives early learning.

**Theorem 1** (Early learning is biased by the OCS mode). *If $\mathbf{1}_N$ is a joint non-degenerate eigenvector to elementwise-positive input and output similarity matrices $\mathbf{X}^T\mathbf{X}$ and $\mathbf{Y}^T\mathbf{Y}$, the OCS mode $s_{ocs}\bar{\mathbf{y}}\bar{\mathbf{x}}^T$ will have leading spectral weight $s_0 \equiv s_{ocs}$ in the SVD of the input-output correlation matrix $\mathbf{\Sigma}^{yx}$.*

*Proof.* Let $\mathbf{1}_N$ be an eigenvector to both similarity matrices $\mathbf{X}^T\mathbf{X}$ and $\mathbf{Y}^T\mathbf{Y}$ associated with eigenvalue $\tilde{\lambda}$. Moreover, let $\mathbf{X}^T\mathbf{X}$ and $\mathbf{Y}^T\mathbf{Y}$ have positive entries. Then, the Perron-Frobenius theorem (Perron, 1907) guarantees that $\tilde{\lambda}$ is indeed the largest eigenvalue (i.e. squared largest singular value) of $\mathbf{Y}^T\mathbf{Y}$, that is: $\tilde{\lambda} \equiv \lambda_0 = s_0^2$. By Proposition 2, the sample-averaged input and output vectors $\bar{\mathbf{x}}$ and $\bar{\mathbf{y}}$ are now also the leading eigenvectors of the "dual" matrices that are contracted along their sample axis, $\mathbf{X}\mathbf{X}^T$ and $\mathbf{Y}\mathbf{Y}^T$. Because the eigenvectors of $\mathbf{Y}\mathbf{Y}^T$ and $\mathbf{X}\mathbf{X}^T$ are the left and right singular vectors of $\mathbf{\Sigma}^{yx}$, respectively, with the eigenvalues being the squares of the singular values, it follows that we can identify the leading eigenmode of $\mathbf{\Sigma}^{yx}$ as

$$s_0\mathbf{u}_0\mathbf{v}_0^T \equiv s_{ocs}\bar{\mathbf{y}}\bar{\mathbf{x}}^T.$$

**Neural tangent kernel** In this section, we review the neural tangent kernel (NTK). This object is useful because it directly describes the learning dynamics in output space $\hat{y}$ (Jacot et al., 2018; Roberts et al., n.d.) as we briefly demonstrate here. We then calculate the NTK for our specific architecture. The following makes use of Einstein summation convention.

For a vector-valued model $\hat{\mathbf{y}}(\mathbf{x}) \in \mathbb{R}^{N_{out}}$ parametrized by a parameter vector $\theta$, the evaluation on sample $\mathbf{x}_i$ from training data at $\mathbf{x}_{i'}$ evolves as

$$\tau\frac{d}{dt}y_m(\mathbf{x}_i) = \sum_k \frac{dy_m(\mathbf{x}_i)}{d\theta^k}\frac{d\theta^k}{dt} \tag{11}$$

$$= -\eta\sum_k \frac{dy_m(\mathbf{x}_i)}{d\theta^k}\frac{d\mathcal{L}}{d\theta^k} \tag{12}$$

$$= -\eta\left[\sum_k \frac{dy_m(\mathbf{x}_i)}{d\theta^k}\frac{dy_{m'}(\mathbf{x}_{i'})}{d\theta^k}\right]\frac{d\mathcal{L}}{dy_{m'}}(\mathbf{x}_{i'}) \tag{13}$$

$$=: -\eta\,\mathsf{NTK}_{mm'}(\mathbf{x}_i, \mathbf{x}_{i'})\left(y_{m'}(\mathbf{x}_{i'}) - \hat{y}_{m'}(\mathbf{x}_{i'})\right), \tag{14}$$

where we used the chain rule and that the parameters update according to gradient descent with learning rate $\eta$, $\frac{d\theta^k}{dt} = -\eta\frac{d\mathcal{L}}{d\theta^k}$. The last line has defined the NTK. We set $\eta = 1$ in the main text for simplicity, as it does not change trajectory and thereby convergence in the case of gradient flow. In addition, we evaluated $\frac{d\mathcal{L}}{dy_{m'}}(\mathbf{x}_{i'})$ for the case of MSE loss $\mathcal{L}(\mathbf{x}_{i'}) = \frac{1}{2}\sum_{m'}\left(y_{m'}(\mathbf{x}_{i'}) - \hat{y}_{m'}(\mathbf{x}_{i'})\right)^2$. The last line of Eq. (11) reveals that the NTK acts as an effective learning rate, as noted by Roberts et al. (n.d.).

We here consider a two-layer linear architecture $\hat{Y}_m^i(\mathbf{X}) = W_{mk}^2\left(W_{kj}^1X_j^i + b_k^1\right) + b_m^2$ where we adopt Einstein summation convention over repeated indices. The parameters are $\theta^k \in \left\{\mathbf{W}^2, \mathbf{W}^2, \mathbf{b}^1, \mathbf{b}^2\right\}$. Herein, $m$ indexes output features and $i$ indexes

data samples. The non-zero gradients are

$$\frac{d\hat{Y}_m^i}{dW_{mk}^2} = W_{kj}^1 X_j^i + b_k^1$$

$$\frac{d\hat{Y}_m^i}{db_m^2} = 1_m$$

$$\frac{d\hat{Y}_m^i}{dW_{kj}^1} = W_{mk}^2 X_j^i$$

$$\frac{d\hat{Y}_m^i}{db_k^1} = W_{mk}^2 1_k.$$

Inserting this into Eq. (11), we get

$$\text{NTK}_{m_1 m_2}(X_j^{i_1}, X_j^{i_2}) = I_{m_1 m_2} \left( X_{j'}^{i_1} W_{j'k}^1 W_{kj''}^1 X_{j''}^{i_2} + b_k^1 b_k^1 \right)$$
$$+ 1_{m_1} 1_{m_2}$$
$$+ W_{m_1 k}^2 W_{km_2}^{2T} X_j^{i_1} X_j^{i_2}$$
$$+ W_{m_1 k}^2 1_k 1_k W_{km_2}^2.$$

or in matrix notation, collecting similar terms

$$\text{NTK}(\mathbf{X}, \mathbf{X}) = \mathbf{I}_{N_{out}} \otimes \mathbf{X}^T \mathbf{W}^{1T} \mathbf{W}^1 \mathbf{X} + \mathbf{b}^{1T} \mathbf{b}^1$$
$$+ \mathbf{1}\mathbf{1}^T \otimes \underbrace{\mathbf{1}\mathbf{1}^T}_{\leftrightarrow \mathbf{b}^2}$$
$$+ \mathbf{W}^2 \mathbf{W}^{2T} \otimes \left( \mathbf{X}^T \mathbf{X} + \underbrace{\mathbf{1}\mathbf{1}^T}_{\leftrightarrow b^1} \right)$$
$$\in \mathbb{R}^{N_{out} \times N_{out}} \otimes \mathbb{R}^{N \times N},$$

where the left hand side operator in the tensor product $\otimes$ is acting in output space $m_1 m_2$, whereas the right hand side operator acts in pattern space $i_1 i_2$. The notation $\leftrightarrow \mathbf{b}$ indicates that a term is due to the bias term. To illustrate this, the NTK acts on the set of labels $\mathbf{Y} \in \mathbb{R}^{N_{in} \times N}$ as follows:

$$(\text{NTK}(\mathbf{X}, \mathbf{X}) \mathbf{Y})_i^m = \sum_{m'}^{N_{out}} \sum_{i'}^{N} \text{NTK}(X_i, X_{i'})^{mm'} Y_{i'}^{m'}. \tag{15}$$

For simplicity, we approximate $\mathbf{W}^2(0) \mathbf{W}^{2T}(0) = \sigma_{\mathbf{W}}^2 I_{N_{out}}$ and $\mathbf{W}^{1T}(0) \mathbf{W}^1(0) = \sigma_{\mathbf{W}}^2 I_{N_{in}}$, which approximately holds for initialization

$$\mathbf{W}^1(0) \sim \mathcal{N}\left(0, \sigma_{\mathbf{W}}^2 / N_{hid}\right), \mathbf{W}^2(0) \sim \mathcal{N}\left(0, \sigma_{\mathbf{W}}^2 / N_{hid}\right), \mathbf{b}^1 = 0, \mathbf{b}^2 = 0$$

where $N_{hid}$ is the size of the hidden layer and both $N_{in}$ and $N_{hid}$ are large. This leaves

$$\text{NTK}(\mathbf{X}, \mathbf{X}) = \sigma_{\mathbf{W}}^2 \mathbf{I}_{N_{out}} \otimes \left( 2\mathbf{X}^T \mathbf{X} + \underbrace{\mathbf{1}\mathbf{1}^T}_{\leftrightarrow b^1} \right) + \mathbf{1}\mathbf{1}^T \otimes \underbrace{\mathbf{1}\mathbf{1}^T}_{\leftrightarrow \mathbf{b}^2}.$$

## Shallow network OCS learning

In this brief section we show the OCS signatures of shallow networks with bias terms. The result is displayed in Fig. 11. We see similar behavioural signatures to deep linear networks. However, the tendency to the OCS is more transient.

## true negative rates in linear networks without bias terms

In this short section we provide a supplemental figure relevant for our results in Section *OCS learning in humans linear networks, and complex models*: We train deep and shallow linear networks *without* bias terms. The learning setting and computation of metrics are equivalent to results in Fig. 4. We display the result in Fig. 12. While networks learn the task, early, response biases are fully absent in these models.

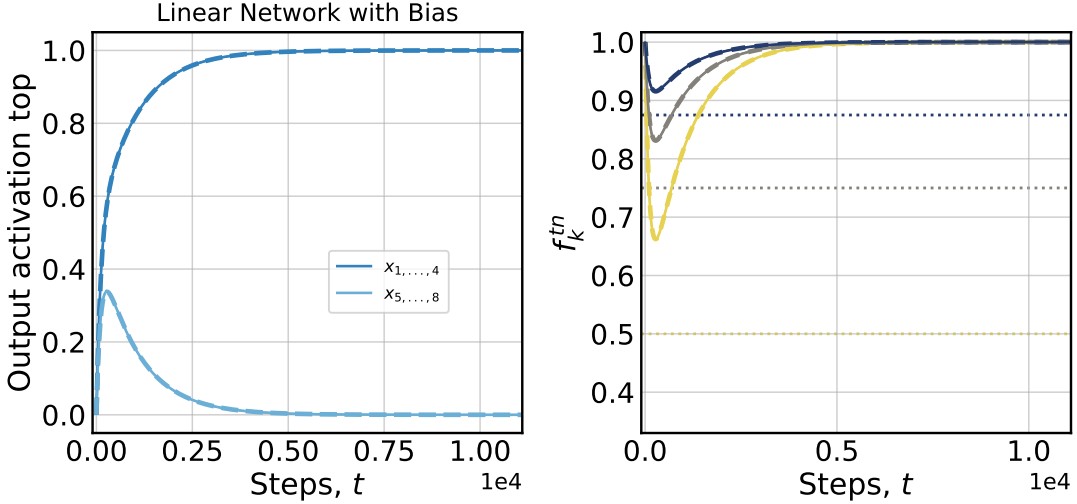

Figure 11: Early learning in shallow networks with bias terms approaches the OCS.

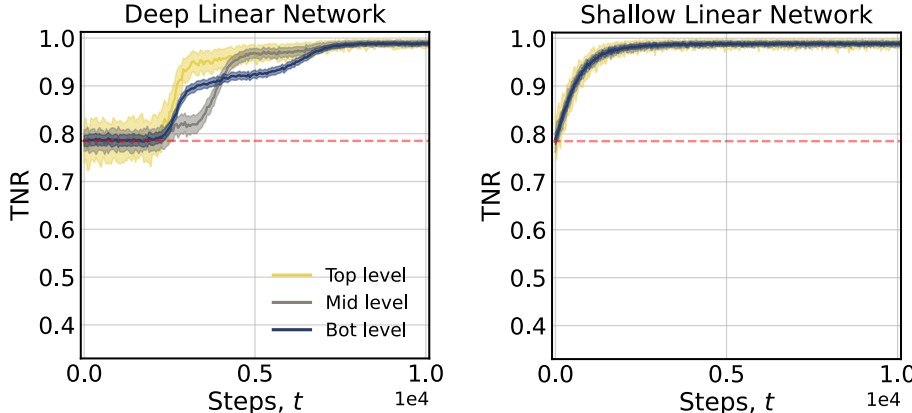

Figure 12: True negative rates for linear networks *without* bias terms. We do not see characteristic response patterns observed in Fig. 4.

## CNN datasets and hyperparameters

**Datasets used.** We used and adapted different image datasets for our experiments with CNNs. While the main text focused on results obtained with a variant of MNIST we report further experiments we conducted to highlight the generality of early OCS learning.

1. **Hierarchical MNIST.** We used the default ten digit classes provided by MNIST. We then sampled 8 classes randomly and replaced the default one-hot labels corresponding to each class $i$ with the hierarchical, "three-hot" labels $y_i$ as seen in Fig. 1. E.g., all images corresponding to MNIST digit "1" might be assigned some random "three-hot" output vector $y_i$.

2. **Hierarchical CIFAR-10.** We applied the same procedure and randomly sampled eight classes from CIFAR-10 and replaced one-hot labels as for the hierarchical MNIST.

3. **Imbalanced-binary-MNIST.** While not described in the main text we also report results in a setting with standard one-hot target vectors. We randomly sample two MNIST classes for training. To assess the impact of the OCS in early learning we introduced class imbalanced by oversampling one of the two classes by a factor of two.

4. **Standard CelebA.** We perform experiments on CelebA's face attribute detection task. The task offers a natural testbed for early learning of the OCS as face attribute target labels form a non-uniform distribution as seen in Fig. 15, bottom. We also normalised images in the dataset before training.

**Model details.** We trained a custom CNN with 3 convolutional layers (layer 1: 32 filters of size 5×5; layer 2: 64 filters of size 3×3; layer 3: 96 filters of size 3×3), followed by 2 fully connected layers of sizes 512 and 256. Activation functions for all layers were chosen as ReLUs. The final layer of the model did not contain an activation function when training with squared error loss. In experiments with the class imbalanced-binary-MNIST and cross-entropy loss the final layer contained a softmax function as non-linearity. For experiments on CelebA the final layer contained sigmoid activation functions and we trained with a binary cross-entropy loss over all 40 labels.

**Training details.** For our results on hierarchical MNIST we train models with minibatch SGD with a batch size of 16 and with a relatively small step size of 1e-4 to examine the early learning phase. For all experiments we used Xavier uniform initialisation (Glorot & Bengio, 2010). Whenever we use bias terms in the model we initialize these as 0 in line with common practice. For our main experiments we train models using a simple squared error loss function. However, to demonstrate generality we repeat experiments for the case of class imbalance using a cross-entropy loss and binary cross-entropy in the case of CelebA. All experiments are repeated 10 times with different random seeds with the exception of CelebA where we used 5 different random seeds, we provide standard errors in all figures (shaded regions). For experiments on the hierarchical CIFAR-10, the class imbalanced MNIST, and CelebA we kept all parameters as above but we increase step size to 1e-3. We trained CNN models on an internal cluster on a single RTX 5000 GPU. Runs took less than one hour to complete.

## Additional experimental results

To understand the generality of OCS learning we plot the results of experiments examining early learning of the OCS in these models. We mostly restrict ourselves to plots as seen in Fig. 5 as we deem these figures most instructive.

**Hierarchical CIFAR-10.** We train on a hierarchical version of CIFAR-10. Where we randomly sample 8 classes from MNIST and replace target labels by hierarchical vectors as in Section *Bias terms drive early learning towards the optimal constant solution*. We find the key signatures of early OCS learning: We find early indifference, the reversion of performance metrics to the OCS, and a small initial distance of average response the to the OCS solution. The results mirror behaviour on the hierarchical MNIST shown in Fig. 3 and Fig. 5.

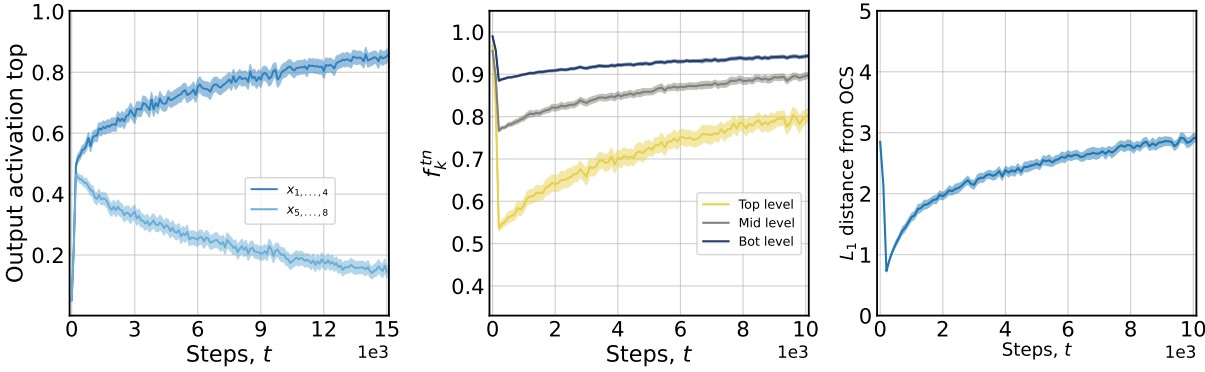

Figure 13: Early OCS learning CNNs trained on hierarchical CIFAR-10. *left:* Network outputs for a single output unit in response to all inputs $\mathbf{x}_i$. *Centre:* Performance metrics $f^{tn}$ (Appendix *True Negative Rates*). *Right:* Mean distance of network responses from OCS. Averages taken over every 10 batches for plotting.

**Class-imbalance MNIST.** We train on an imbalanced MNIST task as described in Appendix *CNN datasets and hyperparameters*. We plot the results for training with squared error and cross-entropy loss functions in Fig. 14. Both settings show reversion to the OCS. Note that average model outputs in the case of the cross-entropy loss start relatively close to outputs expected under the OCS. Despite this proximity the model is still driven towards the OCS solution. The results on this imbalanced case highlight potential fairness implications. Given that network have been found to revert to the OCS when generalising (Kang et al., 2024), early learning in the OCS setting can transiently, but significantly disadvantage minority classes. We further highlight this point in a second solvable case of linear networks with bias terms in Appendix *Linear networks under class imbalance*.

**Standard CelebA.** We show distance from the OCS for the CelebA face attribute detection task in Fig. 15, top. CelebA provides a useful test for our hypothesis as attribute labels display natural imbalances. We highlight the strong non-uniformity of the majority attribute labels in Fig. 15, bottom. We again train networks in two variants: one with squared-error loss and one with binary cross-entropy loss applied over all 40 face attributes. With both loss functions network responses are driven towards the OCS in early learning. This case further highlights the generality of early OCS learning. OCS learning might be especially undesirable in this setting for fairness reasons as the model will be overly liberal in the prediction of the most common face attributes.

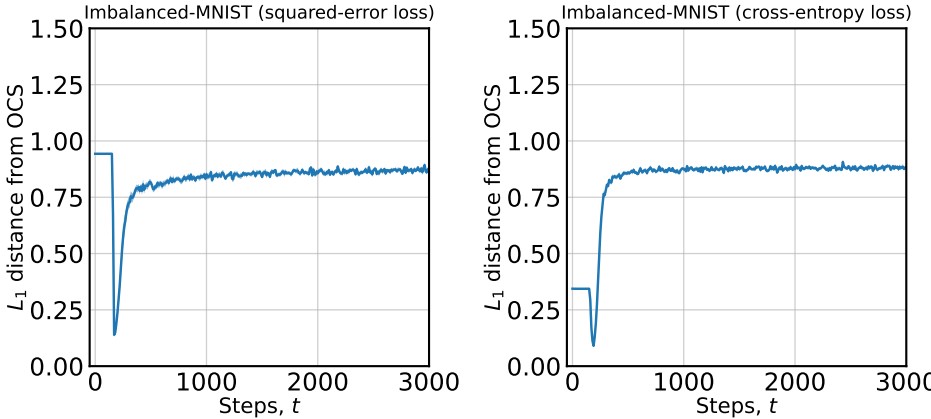

Figure 14: Mean distance of network responses from OCS in CNNs trained on the imbalanced MNIST task. Averages taken over each batch.

## Linear networks under class imbalance

In this section, we describe a second case of a solvable linear network with bias terms. Our dataset consists of two examples where one example appears twice as frequently. We show the data used on the right side of Fig. 16. The minority class has two identifying labels, while this construction appears artificial, it allows for the application of Proposition 1 and solutions to learning dynamics from Section *Linear network preliminaries* apply.

The case is of particular practical relevance as it illustrates the impact of early OCS learning under class imbalance, a common problem in machine learning where datasets are often naturally imbalanced (V. Feldman, 2020; Van Horn & Perona, 2017). In practice, these settings are often addressed through oversampling of minority classes (Haibo He & Garcia, 2009; Huang et al., 2016). Empirical work by Ye et al. (2021) documented that neural networks initially fail to learn information about the minority class while classifying most minority examples as belonging to the majority class. Subsequent theoretical work by Francazi et al. (2023) demonstrated that the phenomenon is caused by competition between the optimisation of different classes.

Our work adds to this literature by providing dynamics in a case of gradient-based learning under class imbalance learning that is exactly solvable. Our exact solutions highlight the potential role of early OCS learning in the initial failure to learn about minority classes. The OCS solution substantially biases early predictions towards the majority class as seen in Fig. 16, centre. The results also can be understood as solvable analogous to early reversion to the OCS seen in the Imbalanced-binary-MNIST setting in Fig. 14. The results highlight the potential fairness implications of early OCS learning as the learning phase systematically biases the model against the minority classes.

## OCS-learning in linear networks with input correlations.

In this section, we demonstrate how input correlations can drive OCS learning in the absence of bias terms in linear networks. Specifically we highlight how OCS learning can emerge if $\mathbf{1}_N$ is an eigenvector of the data similarity matrix $\mathbf{X}^T\mathbf{X}$. Note that the network contains no input correlations. In the bottom row of the Fig. 17, we can see that the first SVD mode $\mathbf{u}_1\mathbf{v}_1^T$ is indeed exactly equivalent to the OCS mode, i.e. $\bar{\mathbf{y}}\bar{\mathbf{x}}^T$. The right panel highlights how the network is driven towards the OCS up until the time-point when the second effective singular value $a_2(t)$ (which is quite close in time) is learned.

## Relation to imbalanced multi-label learning

Given the hierarchical structure of labels used in the majority of our experiments we also see some general connections of our work to problems in the domain of imbalanced multi-label learning. Multi-label learning deals with learning problems in which a single input example is associated with multiple output labels simultaneously. In these settings class imbalance is a key challenge that frequently hinders good performance of models (B. Liu et al., 2020; Charte et al., 2015; Pham et al., 2021; W. Liu et al., 2022). Similar to standard classification problems model biases are frequently addressed through adjustments to the models loss function via selective reweighing (Cui et al., 2019) or through sampling based methods which selectively over- or under-sample particular labels (Charte et al., 2015) or via both methods (Pham et al., 2021). Our results on the hierarchical learning task and on the problem of class imbalanced learning in Appendix *Linear networks under class imbalance* might hint at OCS learning as a potential contributor to problems observed in multi-label learning as the imbalanced distribution of output labels might drive learning to undesirable solutions.

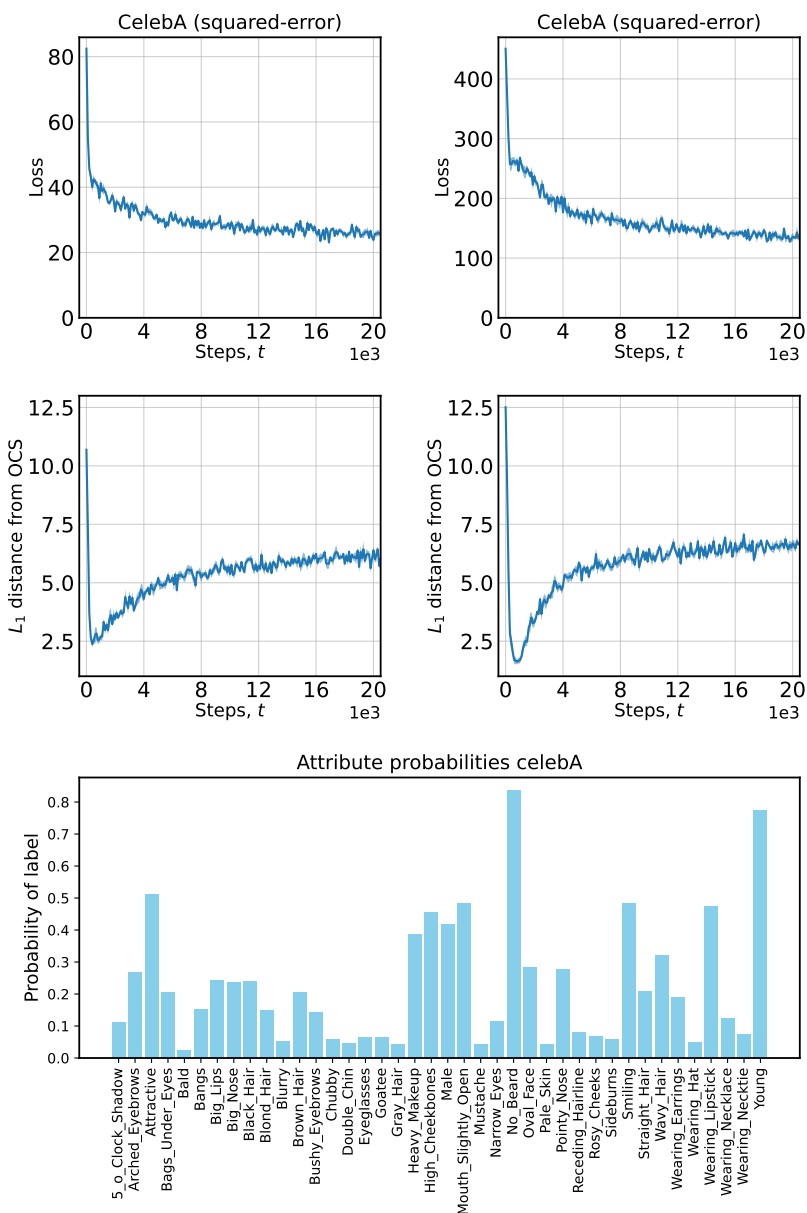

Figure 15: *Top:* Mean distance of network responses from the OCS in CNNs trained on the CelebA face attribute prediction task. *Bottom:* Marginal probabilities of CelebA face attributes.

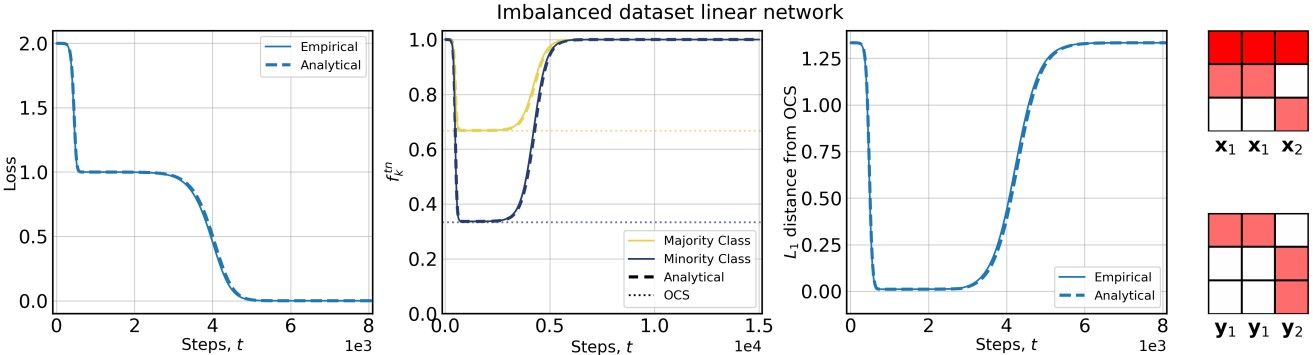

Figure 16: Early learning of the OCS in linear networks under class imbalance.

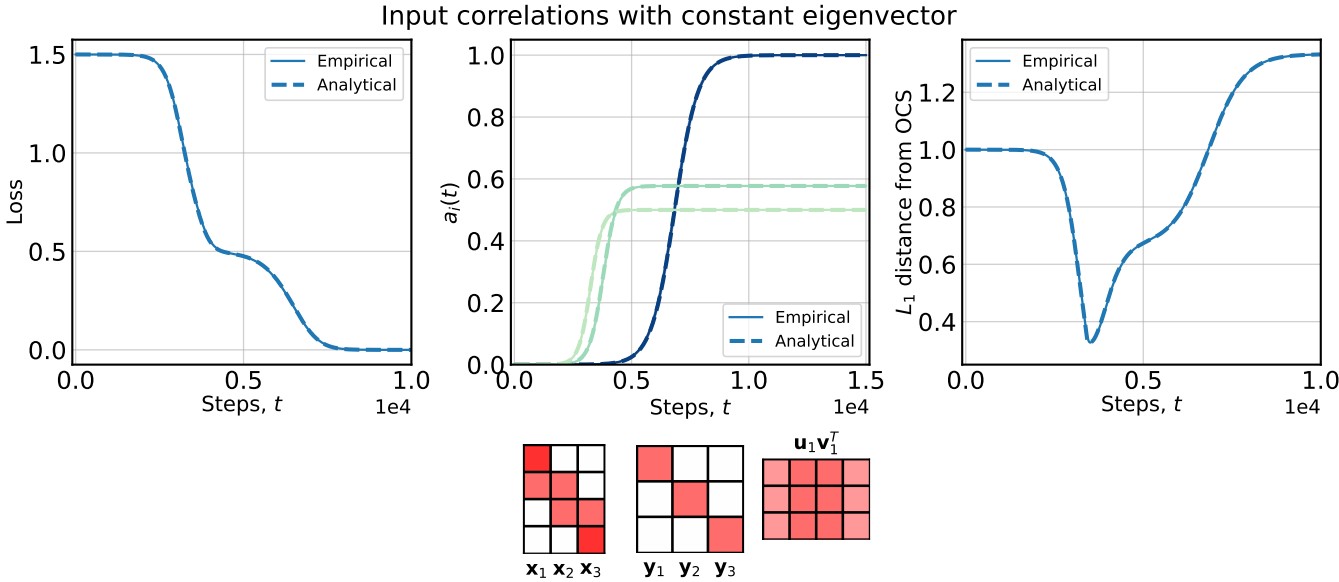

Figure 17: Early learning of the OCS in linear networks with input correlations but in the absence of bias terms.

