# OpenReview forum: "Early learning of the optimal constant solution in neural networks and humans"
_ccneuro.org/CCN/2025/Proceedings — CCN 2025 Proceedings asProceedingsPoster_

### Official Review · Reviewer_tfzD · 2025-03-28
**Strong analysis of a clear and interesting phenomenon of learning systems**

**Soundness:** 3
**Clarity:** 2

**Comments:**

_Summary_: This paper focuses on the phenomenon by which learners (human and artificial) initially learn the dataset output statistics before later learning input-dependent output predictions. This is verified with original human behavioral data alongside empirical studies on nonlinear neural networks. The central contribution of this work is an analytical study of why this happens due to error-corrective learning using the theoretical model system of linear networks. This work extends previous theoretical work in that domain in new ways, being the first to analyze the effect of bias terms.

_Overall evaluation_: This is a very interesting work which exemplifies how analytical analysis of the dynamics of learning in neural networks can lead to interesting qualitative forms of understanding about what learners learn and why. The theoretical work is very thorough. The justification of this work and review of the literature is very strong, and I found it a pleasure to read. The takeaway is quite memorable. I think this will be of interest to many fields.

_Major comments: I have no points of major criticism. Clarity is of some concern; I wonder whether readers with less familiarity with deep linear net analysis will understand the main point. Perhaps leading with the human data might help. I also think that Figure 1 is not very clear unless you've read the paper already and should be reworked.

_Minor comments_:
1. In line 206 it is stated that
> While solutions can be derived for some non-white inputs, little attention has been devoted to learning dynamics in these scenarios.

There is prior work that should be cited that focuses on this setting of non-white inputs. Benjamin et al. 2022 [1] analyze this setting and connect it to human behavior and nonlinear networks (see the Appendix for derivation). Saxe et al. 2019 dedicate a small section of their Appendix as well to this setting for a 2-layer network.

2. I did not quite understand the proof to *Constant data mode is the leading eigenvector*. Please elaborate.
3. The NTK analysis in the Appendix appears extraneous; it is not mentioned in the main body.
4. I found the effect of the output bias terms interesting, yet that was not mentioned in the main text either. I think this should be mentioned. I wonder as well the relative contribution of terms for arbitrary depth N>2 networks; perhaps this may explain the human and CNN data in Fig 4 a bit better.

[1]
@article{benjamin2022efficient,
  title={Efficient neural codes naturally emerge through gradient descent learning},
  author={Benjamin, Ari S and Zhang, Ling-Qi and Qiu, Cheng and Stocker, Alan A and Kording, Konrad P},
  journal={Nature Communications},
  volume={13},
  number={1},
  pages={7972},
  year={2022},
  publisher={Nature Publishing Group UK London}
}

**Expertise:**

3

**Interest:**

3

---

> ### Author Rebuttal · Authors · 2025-04-14
>
> Thank you very much for your review. We are very grateful for the helpful comments and constructive feedback. We are delighted that you found the paper a pleasure to read and that you liked the main takeaway of our work.
>
> *1. There is prior work that should be cited* Thank you for pointing us to these references! We will make sure to add these to the paper. We have uploaded the revised PDF.
>
> *I did not quite understand the proof to Constant data mode is the leading eigenvector…*
> Thanks for this feedback! We edited the proof in the Appendix to clarify the main steps and improved the notation. Let us know if there are any remaining or particular things that are unclear and we will happy to respond and add these details for the final version.
>
> *3. The NTK analysis in the Appendix appears extraneous; it is not mentioned in the main body.* We think it might have gone unnoticed, but we do have small reference to the NTK analysis in the main body of the paper in the very end of the section *Bias terms drive early learning towards the optimal constant solution*. While not strictly needed for our results, we believe that the NTK framing is useful to connect to the wider work on learning dynamics (Roberts 2021, Atanasov 2021).
>
> *4. I found the effect of the output bias terms interesting…* We agree that the bias term on the output layer, with its corresponding exponential learning dynamics, is interesting and might explain why humans are only transiently driven towards the OCS. We will add a more explicit reference in the main text of the paper and will upload a revised pdf.

---

> > ### Comment · Reviewer_tfzD · 2025-04-18
> >
> > Thank you, I think these changes will improve the clarity of the manuscript. Regarding #4, let me add that I missed on my first read that the output layer has no bias in the main analysis (even though it was mentioned), as it's a little less intuitive. It was only upon reading the Appendix that I realized this. So yes, for clarity I think it's great to mention this choice earlier.

---

> > > ### Author Response · Authors · 2025-04-18
> > >
> > > Thank you for your response. We are glad that you think our changes have improved the overall clarity of the paper.

---

### Official Review · Reviewer_yiBs · 2025-03-30
**A rigorous and interesting contribution to the theory of deep learning and cognitive science of learning**

**Soundness:** 3
**Clarity:** 2

**Comments:**

Building on well-known prior work by (Saxe et al.) which elucidated the learning dynamics of deep linear networks and compared them with human learning dynamics on hierarchical datasets, this paper investigates the addition of biases (additive learnable constants) in the layers of deep linear networks, and determines how they alter the (closed-form) learning dynamics of networks. Given biases, and provided that there is an imbalance in the output dim. frequencies (induced by the hierarchical structure of the dataset), the mean prediction for each output dim. will be the first thing to be learned for all output dimensions, i.e. the optimal constant solution (OCS). This phenomenon is also observed in human subjects in experiments provided in the paper (and in prior literature?), suggesting that the learning of biases in early layers could also be at play in human learning. However, in another part of the paper, it is shown that biases are not strictly necessary to see the OCS effect, as it suffices that the constant mode be dominant in the data itself (through shared features across all inputs) for it to be learned first and OCS to appear.

I found the paper to be interesting and fitting regarding two areas relevant to CCN: (1) the dynamics of learning in theory of deep learning and (2) the mechanisms of learning in humans. The importance of the findings is weakened by the fact that one cannot distinguish between two hypotheses regarding human learning (learned biases vs. correlated input features), but the work is still conceptually interesting in that it demonstrates the equivalence of these two hypotheses regarding learning dynamics.

The paper is very sound: the derivations I checked are sound, theory matches experiments, claims are well supported, adequate caveats are mentioned throughout.

The paper is mostly clear, although I have minor suggestions to improve readability (see below). The methods and results are detailed adequately to facilitate reproducibility, and figures illustrate well the concepts and results.

In conclusion, I believe that this is relevant, sound, and clear work that could well deserve publication at CCN.

Questions and suggestions:

- Reading the intro, I was confused about whether the paper investigates the effect of bias terms in input layers, or of a bias in output class frequencies. After reading the paper I understand that it investigates the interaction between both these things. Maybe the distinction and relation between these two concepts could be made more explicit in the intro.

- Shouldn't the equation describing the learning dynamics of the biases (eq 8 currently in appendix) figure more prominently, if it is one of the main results of the paper? This equation could also be compared with the learning dynamics of network weights. Is it fundamentally different or the same?

- It is not totally clear to me whether early OCS learning could happen or not on a dataset which would not present a hierarchical structure. I think it could, but then why even assume a hierarchical dataset to start with?

- In fig 4: What does "Bot level" stand for?

- " 483 Crucially, in linear networks
484  OCS learning hinges on bias terms in the network architecture.
485  However, in non-linear architectures, such as CNNs, the net-
486  work is driven towards the OCS even in the absence of bias "
=> This phrasing is confusing as it suggests that linear networks would not be driven towards the OCS in the absence of bias, although they are if inputs have shared features (i.e. correlated inputs)

- "Response biases seem more transient in
477 shallow networks and appear to more closely mirror human
478 learners."
=> Could one deduce from this that humans are "shallow" learners? It would match anatomy (eg https://www.nature.com/articles/s41583-023-00756-z)

**Expertise:**

2

**Interest:**

2

---

> ### Author Rebuttal · Authors · 2025-04-14
>
> Thank you very much for your thorough review and constructive feedback. We are glad that you found our paper to be interesting and relevant to the CCN community. We are also encouraged by your assessment that our paper is sound and well supported.
>
> *Reading the intro, I was confused…* Thank you we have edited the intro in a few select places and in the contribution section to highlight when we talk about bias terms. We hope that this helps with the readability of the paper. Please let us know if you would like us to rework this section more and we can make additional edits for the final version. We have updated the revised pdf.
>
> *Shouldn't the equation describing the learning dynamics of the biases…?* We decided to omit this equation from the main text of the paper as we absorb the bias term in the first weight matrix of the network $\mathbf{W}^1$ which allows us to use standard solutions for deep linear networks. We found that this treatment strikes the best tradeoff between comprehensiveness and granularity.
>
> *It is not totally clear to me…* You are correct that OCS learning can emerge in non-hierarchical datasets. We highlight such a case in the appendix Linear networks under class imbalance. Our reason to still choose the hierarchical dataset is two-fold. First, it has desirable properties which make it more transparent how the OCS drives early learning, i.e. labels are not uniformly distributed. Secondly, the hierarchical dataset is well studied in connectionism (Rogers & McClelland, 2004; Saxe et al., 2019) and we aimed to build on these results.
>
> *What does "Bot level" stand for?* Bot level stands for the “bottom level” of the output hierarchy.
>
> *This phrasing is confusing* Thanks for the careful reading, we have edited this sentence in the revised pdf to be more suggestive.
>
> *Response biases seem more transient…* We agree that one could conclude this. However, we refrain from making such general statements as we believe that the length of the OCS phase could also be modulated by experimental factors. For instance, we expect that under increased perceptual uncertainty about input-stimuli, i.e. via noisy images, the OCS phase might be accentuated. In such settings, learning the target function is more difficult and learning via label statistics might be the only viable way to reduce error in early learning. We aim to explore these hypotheses in future experiments while here emphasizing causes behind key qualitative signatures.

---

> > ### Comment · Reviewer_yiBs · 2025-04-18
> >
> > _(This is TPC providing a pointer to the [**Official Comment**](https://openreview.net/forum?id=6Xyu486HRh&noteId=nyLMqSjLzB) from Reviewer yiBs starting with:_
> >
> > > Thanks for the clarifications, and ...
> >
> > _With this **Rebuttal Comment** posted on behalf of the Reviewer, the Authors can respond with one final **Reply Rebuttal Comment** during the Author-Reviewer Discussion phase.)_

---

> > > ### Author Response · Authors · 2025-04-18
> > >
> > > Thank you for your response! We are glad that you found our edits to the introduction useful.
> > >
> > > Regarding your question, if the assumptions of Theorem 1 of the main text (and the general assumptions needed to derive exact dynamics in linear networks) are fulfilled then the OCS component should always be learned fastest. In practice, the constant eigenvector for the input-correlation matrix \mathbf{X}^T\mathbf{X} could be induced by either input correlations or bias terms. The statement is general and holds irrespective of other dataset statistics. We hope that this helps to answer your question!

---

### Official Review · Reviewer_frKn · 2025-03-30
**Theoretical account of early phases of learning**

**Soundness:** 3
**Clarity:** 2

**Comments:**

This paper investigates learning in both neural networks and humans. Specifically, they study the “optimal constant solution” (OCS), how networks/human learn as if every input belongs to the distribution’s most common labels, ignoring input-specific distinctions. The paper shows that bias terms are critical for OCS learning, and more broadly, provides insight into strategies devised during early phases of learning.

The paper is generally clearly written, and the figures are helpful. The study is very comprehensive: analyzing several network architectures as well as human behavior--making it a great fit for the CCN audience. One limitation is that the paper seems to be suited better for a longer form than just 8 pages--many of the Appendices are quite critical for the paper. Also, the paper is not very simple to read for non-experts in the domain, and in places it could be more clear why the authors perform the analyses they do, and what they predict the outcomes will be. Finally, my main comment on limitations is that it is hard to understand exactly how the current work is novel from prior work. The authors do a great job of citing prior work, but sometimes the boundary between prior work and the current work is not clear, especially for readers who are not very familiar with the prior literature.


Smaller comments / questions:
- It would be helpful if the authors could spell out more explicitly what they mean when they introduce "importance", e.g., L52?
- What do the authors mean by "output statistics of the data" (L63)?
- It would be helpful to have color labels and sub-headers in Figure 2.
- It would be helpful to have a bit more about the human behavioral experiment in the main text (the Appendix is helpful)--but the human experiment is a bit hard to contextualize without looking at Appendix, such as number of participants, general task structure etc.
- Why do humans show a much smaller drop in true negative rate compared to the different networks, Figure 4?

**Expertise:**

1

**Interest:**

3

---

> ### Author Rebuttal · Authors · 2025-04-14
>
> Thank you for your thoughtful comments and feedback. We're glad you found our work comprehensive, clearly written, and a good fit for the CCN community given our approach spanning neural network and human experiments.
>
> We recognize the paper's density and frequent references to the Appendix, but this reflects our attempt to balance providing technical details while meeting CCN's space constraints. We believe CCN remains uniquely suitable for our interdisciplinary work.
>
> Regarding accessibility, we included essential details about the linear network formalism in the introduction to make the paper relatively self-contained, and hope that the main statement—that early learning is driven towards the OCS—is emphasized even without understanding all technical aspects.
>
> Thank you for your feedback on distinguishing from existing literature! In the Related works section, we now added a contrasting sentence that highlights our contribution to each line of prior work.
>
> #### Smaller comments
>
> *importance?* Importance in this context refers to the magnitude of singular values in the SVD on the input-output correlation matrix. Similarly to other dimensionality reduction techniques (e.g. automatic-relevance detection) the magnitude of the singular values will capture how much information about the matrix is captured by associated singular vectors.
>
> *output statistics of the data?* We mean statistics of the labels, i.e. the average output vector $\bar{\mathbf{y}}\in \mathbb{R}^{N_{out}}$
>
> *Details on human experiment* We have added some additional detail to the human experiments and have updated the PDF. However, due to space constraints we will keep most details in the Appendix.
>
> *Why do humans show a smaller drop?* We can only speculate about the exact origin of the difference. One noteworthy observation is that humans behave quite similarly to shallow networks. However, we also believe that the exact duration of the initial OCS phase in humans might be modulated by experimental factors. We expect that under increased perceptual uncertainty about input-stimuli, i.e. via noisy images, the OCS phase might be accentuated. In such settings, learning might be more difficult and learning via label statistics might be a more viable way to reduce error. While our focus is on the qualitative driving factors of the OCS, we aim to explore these hypotheses in future experiments.

---

### Meta-Review · Area_Chair_uagc · 2025-05-04

**Ccn Recommendation:** Accept as Proceedings

**Metareview:**

The authors effectively addressed all reviewer concerns regarding clarity for non-experts, distinction from prior work, and technical explanations. The paper demonstrates methodological rigor by combining theoretical analysis with empirical validation across both neural networks and human subjects. I recommend acceptance of this paper to the proceedings as it makes a significant contribution by theoretically analyzing optimal constant solution learning, and is supported by strong empirical evidence from both neural networks and human participants.

**Summary:**

The authors investigated the optimal constant solution of learning for both neural networks and humans. The reviewers were in consensus that this is a high quality paper with valuable insights for biology and AI, and which has strong empirical support and interdisciplinary relevance. The reviewers also praised the comprehensive approach and indicated there were novel contributions for analyzing learning dynamics. Concerns were minimal but focused on the clarifications/potentially some dense writing/lack of differentiation, and these were largely addressed during the rebuttal period.

**Expertise:**

2